# DYNAMIC LOCALLY LINEAR GRAPH LEARNING FOR GEOMETRY-AWARE GNNS

## ABSTRACT

The accuracy of GNN-based node classification depends critically on the quality of the constructed graph. Common heuristics such as $k$NN or Gaussian kernels rely on Euclidean distances, which in high dimensions fail to capture manifold structure and often introduce spurious cross-class edges. During message passing, these edges propagate inconsistent signals and, with deeper layers, lead to over-smoothing where node embeddings become indistinguishable. To address these issues, we propose VLGNN, an end-to-end framework that integrates graph construction with GNN training. A variational autoencoder (VAE) encodes the full data distribution into a latent space, thereby capturing global structure, while an LLE-inspired module refines the graph by enforcing local manifold consistency. The graph structure is updated jointly with the VAE-based encoder and the GNN-based classifier, allowing neighborhoods to adapt to the learned representations and to preserve meaningful within-class relations during training. Spectral analysis further shows that our LLE-based method enlarges Laplacian eigengaps and reduces inter-class conductance, indicating weaker cross-class propagation and alleviation of over-smoothing. On standard benchmarks, VLGNN achieves higher accuracy with reduced cross-class mixing.

## 1 INTRODUCTION

Graph neural networks (GNNs) have become powerful tools for learning from relational data, achieving state-of-the-art results in tasks such as semi-supervised node classification and recommendation (Gao et al., 2023). Their effectiveness, however, relies heavily on the quality of the underlying graph: edges govern information flow, neighborhood interactions, and ultimately label inference (Shi et al., 2023). In many real-world scenarios, data are not natively graph-structured—sensor signals, feature vectors, or images must first be transformed into a graph before GNNs can operate. This construction step is therefore critical, as it determines whether representations encode meaningful relationships or are corrupted by noisy, misleading connections (Zhao et al., 2024; Ju et al., 2024). Most existing approaches build graphs heuristically (e.g., $k$-NN or cosine similarity), a practice still common in recent systems and graph structure learning frameworks (Tamaru et al., 2024; Tenorio et al., 2024; Ruys et al., 2025). Such methods often ignore the intrinsic geometry of high-dimensional data on low-dimensional manifolds, leaving them sensitive to noise and prone to spurious cross-manifold edges that destabilize message passing and degrade performance (Wu et al., 2023a). Even with a reasonable base graph, deeper propagation can still homogenize node features (over-smoothing), further eroding discriminative power (Rusch et al., 2023).

While many techniques mitigate the limitations of deep GNNs—such as residual connections, normalization layers, personalized propagation, and spectral rewiring—they primarily operate at the architectural or message-passing level (Scholkemper et al., 2025; Karhadkar et al., 2022). These methods preserve feature variance and stabilize training but rely on a fixed or heuristically modified graph, leaving the underlying geometric structure unmodeled (Wang et al., 2025; Wei et al., 2025). From a manifold perspective, data typically reside on locally linear patches, suggesting that graph edges should encode affine relationships rather than arbitrary pairwise similarities (Meilă & Zhang, 2024). Locally Linear Embedding (LLE) provides such a mechanism by reconstructing each sample as an affine combination of its neighbors, thereby preserving local geometry and suppressing spurious edges (Xue et al., 2023). Incorporating this geometric prior into graph construction yields more faithful neighborhood structures and mitigates over-smoothing (Oono & Suzuki, 2021). How-

ever, LLE has traditionally been used as a preprocessing step, with the graph remaining static during training. Recent studies emphasize that jointly optimizing structure and representation is crucial for robust and expressive GNNs—yet this remains underexplored (Zhou et al., 2023a).

Motivated by these observations, we propose an end-to-end semi-supervised framework that integrates a variational autoencoder (VAE) (Kingma & Welling, 2013), LLE-based dynamic graph construction, and a GNN classifier. The VAE learns latent representations in which local neighborhoods are encouraged to be linearly reconstructible; LLE derives sparse, geometry-faithful weights in this latent space to update the graph; and the GNN performs label propagation on the evolving structure. By optimizing these components together, the graph and node representations are jointly optimized to encourage locally affine neighborhoods and discriminative boundaries, effectively combining geometric priors with deep representation learning.

The main contributions of this work are summarized as follows:

- **Geometry-guided dynamic graph construction.** We propose an end-to-end framework that couples VAE embeddings with LLE-based dynamic graph updates, thereby incorporating a local affine prior into graph structure learning.

- **Theoretical insights into depth stability.** We theoretically analyze how LLE-based reconstruction tends to suppress spurious cross-class edges and approximately preserve local geometry. From a spectral perspective, our analysis further suggests that the induced Laplacian redistributes energy across frequencies, which can help moderate spectral homogenization and thereby mitigate over-smoothing.

- **Robust empirical improvements.** On multiple sensor-based datasets with limited labels, our method consistently improves classification accuracy, with spectral analysis suggesting enhanced stability when training deeper GNNs.

- **Interpretability and simplicity.** Our graph updates are weight-space transparent (explicit LLE coefficients), computationally efficient, and readily compatible with standard GNN architectures.

## 2 RELATED WORK

**Semi-Supervised GNNs and Over-Smoothing.** Graph neural networks (GNNs) such as GCN (Kipf & Welling, 2017) and GAT (Veličković et al., 2018) have proven effective for semi-supervised node classification. However, increasing depth often leads to *over-smoothing*, where node embeddings converge to indistinguishable representations (Oono & Suzuki, 2021). Related issues such as *over-squashing* further hinder long-range information flow (Alon & Yahav, 2021). Remedies include residual connections (Xu et al., 2018), normalization (Zhao & Akoglu, 2020), and personalized propagation (Klicpera et al., 2019b), which stabilize training of deeper GNNs. Yet, these methods are largely architectural fixes and do not explicitly encode the geometric manifold structure of the input space. This raises the question of whether incorporating geometric priors can further mitigate such limitations.

**Graph Structure Learning.** Another research line learns or refines graph structures jointly with GNN training (Franceschi et al., 2019; Zhu et al., 2022). While recent approaches improve robustness via entropy, mutual information, or feature similarity criteria (Zou et al., 2023; Zhou et al., 2023b), they typically assume a static graph or rely on heuristic priors. More recent work has considered dynamic graphs in specific domains, e.g., DBGSL for dynamic brain graph learning (Campbell et al., 2024) and DG-Mamba for spatio-temporal settings (Yuan et al., 2025). Despite these advances, most methods still depend on implicit heuristics and do not explicitly enforce local manifold structure.

**Manifold Methods and LLE.** Separately, the manifold learning literature has developed geometry-preserving embedding methods such as Locally Linear Embedding (LLE) (Roweis & Saul, 2000) and Laplacian Eigenmaps (Belkin & Niyogi, 2003). These approaches assume data lie on a low-dimensional manifold and construct graphs that preserve local neighborhood relations. They have been widely used for unsupervised learning and visualization, but are typically applied as preprocessing steps rather than being integrated into end-to-end GNN frameworks. A few attempts

combine manifold priors with deep networks (Jayapal & Annamalai, 2025), but such approaches remain rare. This gap motivates our method, which injects LLE-inspired manifold priors directly into dynamic graph updates to enhance GNN training. Additional details on over-smoothing analyses, graph structure learning frameworks, and manifold methods are provided in Appendix A.

# 3 PROPOSED METHOD - VLGNN

## 3.1 PRELIMINARIES

**Variational Autoencoder (VAE).** A VAE (Kingma & Welling, 2013) maps an input $x_i$ into a latent code $z_i$ via an encoder $q_\phi(z_i|x_i) = \mathcal{N}(\mu_i, \text{diag}(\sigma_i^2))$ and reconstructs it through a decoder $p_\theta(x_i|z_i)$. The objective combines a reconstruction term and a KL regularization with respect to a prior $p(z_i) = \mathcal{N}(0, I)$:

$$\mathcal{L}_{\text{VAE}} = \sum_i \mathbb{E}_{q_\phi(z_i|x_i)}[-\log p_\theta(x_i|z_i)] + \beta \, \text{KL}(q_\phi(z_i|x_i) \,\|\, p(z_i)), \tag{1}$$

where $\beta$ is an optional weighting factor as in $\beta$-VAE.

**Locally Linear Embedding (LLE).** Given latent features $z_i$, LLE (Roweis & Saul, 2000) reconstructs each point as a linear combination of its neighbors:

$$z_i \approx \sum_{j \in \mathcal{N}(i)} w_{ij} z_j, \quad \text{s.t.} \sum_j w_{ij} = 1, \tag{2}$$

where $\mathcal{N}(i)$ represent the neighbor set of size $k$ for node $i$, and $w_{ij}$ are obtained by the least-squares optimization. The weights induce the following matrix

$$M = (I - W)^\top (I - W), \tag{3}$$

which is closely related to the Laplace–Beltrami operator (Ting et al., 2010), thereby preserving local geometric structure.

**Graph Neural Networks (GNNs).** Given an adjacency matrix $A$, a GNN layer (Kipf & Welling, 2017) updates hidden features as

$$H^{(\ell+1)} = \sigma(\hat{A} H^{(\ell)} W^{(\ell)}), \tag{4}$$

where $\hat{A} = D^{-\frac{1}{2}}(A + I)D^{-\frac{1}{2}}$ is a normalized adjacency matrix with self-loops, $\sigma$ is a nonlinearity, and $W^{(\ell)}$ are learnable weights.

## 3.2 MOTIVATION

The insights from Section 2 suggest that **geometry-aware graph construction** can complement architectural remedies. If graph edges are encouraged to reflect local affine combinations, the resulting Laplacian becomes more aligned with manifold geometry and can provide more faithful neighborhoods for label propagation. However, in practice, node features may not naturally reside in a space where the LLE assumptions hold. To address this, we propose to learn latent representations with a variational autoencoder (VAE), which encourages neighborhoods to be more linearly reconstructible (Netto et al., 2025). Based on these latent codes, we dynamically construct an LLE-inspired adjacency matrix and then propagate labels with a GNN. This design explicitly incorporates manifold priors into graph structure learning, while coupling them with end-to-end representation learning.

**Framework Overview.** Our framework integrates three components: (i) a variational autoencoder (VAE) that maps raw features into a latent space where local neighborhoods are encouraged to be linearly reconstructible, (ii) an LLE-based module that constructs a geometry-preserving adjacency matrix from the latent codes, and (iii) a graph neural network (GNN) that performs semi-supervised classification on the dynamically updated graph. These components are trained jointly in an end-to-end manner, so that node embeddings and graph structure co-evolve throughout optimization. Figure 1 provides an overview of the pipeline.

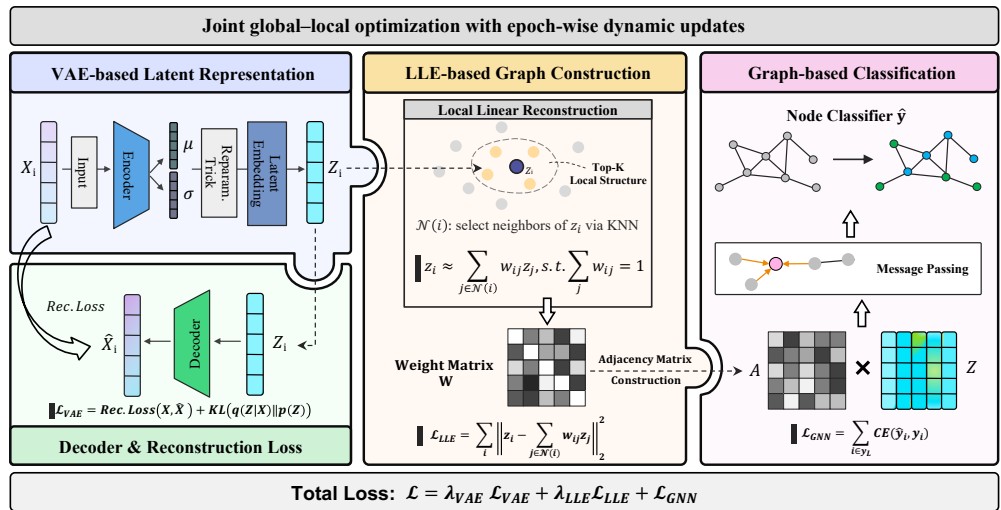

Figure 1: Overview of **VLGNN**, our proposed framework.

**Joint Objective.** The training objective integrates three complementary terms:

$$\mathcal{L} = \mathcal{L}_{\text{GNN}} + \lambda_{\text{VAE}}\mathcal{L}_{\text{VAE}} + \lambda_{\text{LLE}} \sum_{i=1}^{n} \left\| z_i - \sum_{j \in \mathcal{N}(i)} w_{ij} z_j \right\|_2^2. \tag{5}$$

**GNN term.** $\mathcal{L}_{\text{GNN}}$ is the cross-entropy loss computed only on the labeled nodes, guiding the GNN towards discriminative predictions under the semi-supervised setting. **VAE term.** $\mathcal{L}_{\text{VAE}}$ combines a reconstruction loss with a KL divergence, encouraging the encoder to learn globally smooth latent codes rather than overfitting to the labeled subset. **LLE term.** The reconstruction residual penalizes deviations from local affine combinations, ensuring that neighborhoods in the latent space remain geometrically faithful and reducing spurious cross-class connections. The coefficients $\lambda_{\text{VAE}}, \lambda_{\text{LLE}}$ balance these auxiliary terms. During optimization, gradients are propagated through the encoder, decoder, and GNN parameters, while the LLE weights are recomputed at each epoch and not treated as trainable parameters. This joint design encourages latent smoothness, geometric consistency, and discriminative power simultaneously.

**Algorithmic Workflow.** Algorithm 1 summarizes the overall training workflow. At each epoch, the encoder of the variational autoencoder (VAE) maps raw node features into a latent representation that is regularized through reconstruction and KL divergence objectives. On these latent codes, locally linear embedding (LLE) weights are recomputed in closed form, producing an adjacency matrix that reflects affine neighborhood structure. This adjacency matrix is further processed by nonnegativity projection, symmetrization, and top-$k$ sparsification to yield a normalized graph that remains sparse and numerically stable. A graph neural network (GNN) then propagates information over this evolving graph, using the latent embeddings as input features. The training objective combines three terms: supervised cross-entropy on the labeled nodes, the standard VAE loss, and a reconstruction residual from the LLE module. Gradients are backpropagated only through the encoder, decoder, and GNN parameters, while the LLE weights are recomputed at each epoch and are not treated as learnable parameters. This alternating procedure ensures that latent representations, graph structure, and predictive layers co-evolve throughout training.

### 3.3 Geometric and Spectral Effects of LLE Graphs

Our method leverages locally linear embedding (LLE) to construct geometry-aware graphs. This design impacts graph structure at three levels: (i) suppressing spurious cross-class connections, (ii) balancing spectral energy for stable propagation, and (iii) mitigating oversmoothing in deep GNNs. We outline these effects below and provide quantitative evidence in Section 4.

---

**Algorithm 1** End-to-end training of VLGNN

---

**Require:** Node features $X$, labeled set $\mathcal{Y}_L$; hyperparameters $(k,\ \lambda_{\text{VAE}},\ \lambda_{\text{LLE}},\ \beta,\ \gamma,\ \epsilon)$; epochs $T$; GNN depth $L$

**Ensure:** Trained encoder/decoder $(\phi, \theta)$ and GNN parameters

1: Initialize $(\phi, \theta)$ and GNN weights $\{W^{(\ell)}\}$
2: **for** epoch $= 1$ to $T$ **do**
3:      **VAE encoding:** Encode $X$ into latent embeddings $Z = \{z_i\}$ with $\text{Enc}_\phi$; reconstruct $\hat{X} \leftarrow \text{Dec}_\theta(Z)$
4:      **Compute LLE weights (closed-form, not trainable):**
5:      **for** each node $i$ **do**
6:         Find $k$ nearest neighbors $\mathcal{N}(i)$ in $Z$
7:         Form local *Gram* matrix $C(i)$ with $C_{jj'}(i) = (z_i - z_j)^\top (z_i - z_{j'})$
8:         Stabilize: $C(i) \leftarrow C(i) + \epsilon I$
9:         Compute weights $W = [w_{ij}]$: $w_{i*} \leftarrow \frac{C^{-1}(i)\mathbf{1}}{\mathbf{1}^\top C^{-1}(i)\mathbf{1}}$, $\sum_{j \in \mathcal{N}(i)} w_{ij} = 1$, $w_{ij} = 0$ if $j \notin \mathcal{N}(i)$
10:      **end for**
11:      Nonnegativity & renormalization: $W \leftarrow \text{ReLU}(W)$; $w_{i*} \leftarrow w_{i*}/\sum_j w_{ij}$ for each row $i$
12:      Sparsify $A_{\text{raw}}$ by keeping top-$k$ entries per row and symmetrize $A_{\text{raw}}$ again
13:      Add self-loops and normalize: $\tilde{A} \leftarrow A_{\text{raw}} + \gamma I$, $\hat{A} \leftarrow D^{-\frac{1}{2}} \tilde{A} D^{-\frac{1}{2}}$, $D = \text{diag}\!\left(\tilde{A}\mathbf{1}\right)$
14:      **GNN forward:** $H^{(0)} = Z$; $H^{(\ell+1)} = \sigma(\hat{A} H^{(\ell)} W^{(\ell)})$, $\ell = 0, \ldots, L-1$; $\hat{Y} = \text{Classifier}(H^{(L)})$
15:      **Compute losses:**

$$\mathcal{L}_{\text{GNN}} = \sum_{i \in \mathcal{Y}_L} \text{CE}(\hat{y}_i, y_i),$$

$$\mathcal{L}_{\text{LLE}} = \sum_i \left\| z_i - \sum_{j \in \mathcal{N}(i)} w_{ij} z_j \right\|_2^2 \quad \text{// gradients only to } Z,$$

$$\mathcal{L}_{\text{VAE}} = \sum_i \mathbb{E}_{q_\phi}\big[ -\log p_\theta(x_i|z_i)\big] + \beta\,\text{KL}\big(q_\phi(z_i|x_i) \,\|\, p(z)\big),$$

$$\mathcal{L} = \mathcal{L}_{\text{GNN}} + \lambda_{\text{VAE}}\mathcal{L}_{\text{VAE}} + \lambda_{\text{LLE}}\mathcal{L}_{\text{LLE}}.$$

16:      **Update only** $(\phi, \theta, \{W^{(\ell)}\})$ by backpropagation
17: **end for**

---

**Local Geometry and Cross-Class Suppression.** LLE reconstructs each point as an affine combination of its neighbors, minimizing local reconstruction error under a linearity prior (Roweis & Saul, 2000). Neighbors from different classes typically lie outside the tangent patch, so including them increases the error and they naturally receive smaller weights. This suppresses cross-class edges that distance-only $k$NN graphs often introduce.

To quantify this effect, we adopt two neighborhood purity metrics:

**Inter-Class Edge Ratio (ICER).** Following prior work (Zhao et al., 2020), ICER measures the proportion of edges that connect nodes from different classes. Starting from the normalized LLE weight matrix $P \in \mathbb{R}^{n \times n}$, we construct a symmetrized, sparsified adjacency $A$:

$$A_{ij} = \mathbf{1}\left[\frac{P_{ij} + P_{ji}}{2} > \tau\right], \quad A_{ij} = A_{ji}, \; A_{ii} = 0. \tag{6}$$

ICER is then defined as:

$$\text{ICER} = \frac{\sum_{i<j} A_{ij}\,\mathbf{1}[y_i \neq y_j]}{\sum_{i<j} A_{ij}}. \tag{7}$$

Lower ICER indicates neighborhoods with fewer cross-class edges.

**Cross-Class Weight Share (CCWS).** ICER ignores edge weights. To capture weight distribution, we define CCWS. Let $W \in \mathbb{R}^{n \times n}$ be the LLE weight matrix, and define its nonnegative part

$$W_{ij}^+ = \max(W_{ij}, 0). \tag{8}$$

Row-normalizing yields

$$P_{ij} = \frac{W_{ij}^+}{\sum_k W_{ik}^+}, \quad \sum_j P_{ij} = 1. \tag{9}$$

CCWS is the average fraction of weight assigned to cross-class neighbors:

$$\text{CCWS} = \frac{1}{n} \sum_{i=1}^n \sum_{j=1}^n P_{ij} \, \mathbf{1}[y_i \neq y_j]. \tag{10}$$

Lower CCWS implies that reconstruction weights concentrate on same-class neighbors.

**Spectral Properties and Energy Balance.** From a spectral perspective, over-smoothing in GNNs is linked to Laplacian eigenvalues concentrating near zero, which accelerates feature homogenization (Oono & Suzuki, 2021). The LLE-induced Laplacian alleviates this issue by enforcing locally affine weights, which approximate the manifold Laplacian and redistribute spectral energy. This suppresses high-frequency leakage and balances low- and high-frequency components.

We evaluate spectral balance with three indicators. Let $L = I - D^{-1/2}AD^{-1/2}$ be the normalized Laplacian of adjacency $A$ with eigenvalues $\{\lambda_i\}_{i=1}^n$. We define:

$$\text{LowFreq} = \frac{\sum_{i \leq 0.1n} \lambda_i^2}{\sum_{i=1}^n \lambda_i^2}, \quad \text{HighFreq} = \frac{\sum_{i \geq 0.9n} \lambda_i^2}{\sum_{i=1}^n \lambda_i^2}, \tag{11}$$

$$\text{Range} = \lambda_{\max} - \lambda_{\min}. \tag{12}$$

Balanced spectra (lower LowFreq/HighFreq, larger Range) indicate delayed oversmoothing and more stable propagation.

**Depth Stability and Oversmoothing Mitigation.** As a result of geometry- and spectrum-aware design, node representations homogenize more slowly, allowing deeper GNN layers to be stacked without immediate collapse. This retains discriminative power across greater depths.

We quantify depth stability with a layer-wise cosine similarity metric (Jiang et al., 2025):

$$\text{Sim}^{(\ell)} = \frac{1}{n} \sum_{i=1}^n \frac{h_i^{(\ell)} \cdot h_i^{(\ell+1)}}{\|h_i^{(\ell)}\| \|h_i^{(\ell+1)}\|}, \tag{13}$$

where $h_i^{(\ell)}$ is the embedding of node $i$ at layer $\ell$. Lower similarity indicates stronger feature change between layers, while higher similarity reflects oversmoothing. LLE-based graphs achieve slower growth of similarity, implying delayed oversmoothing (see Sec. 4).

## 4 EXPERIMENTS

### 4.1 DATASETS

We evaluate our method on three wearable-sensor benchmarks for human activity recognition (HAR). **HAR** (Reyes-Ortiz et al., 2013) contains 6 daily activities with 561 handcrafted features extracted from smartphone accelerometers (10,299 samples). **HAPT** (Reyes-Ortiz et al., 2015) extends HAR to 12 activities including postural transitions (10,929 samples with the same 561 features). **WISDM** (Weiss, 2019) contains 6 activities collected in unconstrained real-world settings, represented by 54 statistical features. Following the official protocol, we use 70% of the data for training and 30% for testing, and further hold out 10% of the training portion for validation. Within the training set, only 20% of the nodes are treated as labeled, while the remaining 80% are unlabeled. For WISDM, we additionally augment the training set with 15,000 unlabeled samples randomly drawn from the provided 1.3M unlabeled pool. All features are standardized per dimension before training. This setting ensures that semi-supervised learning is evaluated fairly, with validation and test sets strictly held out for model selection and final evaluation.

Table 1: Node classification accuracy (%) on HAR, HAPT, and WISDM benchmarks. We compare our proposed dynamic LLE-based graph construction with a range of baselines, including (i) *static heuristics* (KNN, Gaussian, Cosine, Correlation, A-KNN) and (ii) *learnable or diffusion-based methods* (IDGL, GDC), all under the same VAE+GCN backbone. Results are reported as mean $\pm$ standard deviation (%).

| Method | HAR | HAPT | WISDM |
|---|---|---|---|
| *Static heuristics* | | | |
| VAE + KNN + GCN (static)  (Bernhardsson, 2018) | $83.6 \pm 0.16$ | $81.3 \pm 0.16$ | $68.0 \pm 0.22$ |
| VAE + Cosine + GCN (static)  (Chen et al., 2020) | $78.2 \pm 0.22$ | $77.8 \pm 0.43$ | $65.1 \pm 0.10$ |
| VAE + A-KNN + GCN (static)  (Yang et al., 2023) | $86.7 \pm 0.42$ | $88.3 \pm 0.50$ | $72.0 \pm 0.32$ |
| VAE + Correlation + GCN (static)  (Cao et al., 2024) | $84.4 \pm 0.20$ | $84.6 \pm 0.32$ | $70.7 \pm 0.34$ |
| VAE + Gaussian + GCN (static)  (Netto et al., 2025) | $93.3 \pm 0.18$ | $92.9 \pm 0.01$ | $76.4 \pm 0.13$ |
| *Learnable / diffusion-based graphs* | | | |
| VAE + GDC + GCN (diffusion)  (Klicpera et al., 2019a) | $93.8 \pm 0.14$ | $93.7 \pm 0.12$ | $80.2 \pm 0.23$ |
| VAE + IDGL + GCN (dynamic)  (Chen et al., 2020) | $94.2 \pm 0.30$ | $93.6 \pm 0.18$ | $80.8 \pm 0.21$ |
| **VAE + LLE + GCN (dynamic) [Ours]** | $\mathbf{95.2 \pm 0.23}$ | $\mathbf{94.3 \pm 0.02}$ | $\mathbf{82.2 \pm 0.04}$ |

**Baselines.** We evaluate our approach against a comprehensive set of graph construction baselines, including both traditional heuristic methods and more advanced learnable or diffusion-based approaches. Specifically, we consider the following static heuristics: (i) $k$-nearest neighbors (KNN), (ii) Cosine similarity graphs, (iii) Adaptive $k$NN (A-KNN), (iv) Correlation-based graphs, and (v) Gaussian kernel graphs. These methods construct a fixed adjacency matrix based on latent representations produced by the VAE, after which a GCN is applied for node classification. This design isolates the effect of graph construction while keeping the encoder and classifier architecture unchanged. For a fair comparison, we tune $k$ (for KNN and A-KNN), $\sigma$ (for Gaussian), and corresponding hyperparameters for Cosine and Correlation graphs using the validation set. Beyond static heuristics, we also compare with two representative learnable or diffusion-based approaches: (vi) GDC, which refines the graph structure through personalized PageRank diffusion; and (vii) IDGL, which jointly learns both the graph structure and node representations. Unlike these baselines, our proposed method dynamically updates the adjacency structure during training by coupling VAE representations with LLE-based local geometry. This enables the graph topology and node embeddings to co-evolve, yielding more faithful neighborhood structures. Additional results with alternative GNN backbones (e.g., SimpleGNN) are reported in Appendix E.

**Overall Accuracy.** Table 1 summarizes node classification performance on HAR, HAPT, and WISDM. Across all three benchmarks, our LLE-based dynamic graph construction achieves the highest accuracy under the same VAE+GCN backbone. Compared with conventional static heuristics (KNN, Gaussian, Cosine, Correlation, A-KNN), our approach yields substantial improvements. On HAR and HAPT, where sensing conditions are relatively stable, the gains are moderate yet consistent: we outperform the Gaussian baseline by $1.9\%$ and $1.4\%$, and achieve larger margins of $11.6\%$ and $13.0\%$ over KNN. On the more challenging WISDM dataset—where signals are collected in unconstrained real-world settings, resulting in higher intra-class variability and stronger cross-class mixing—our method improves over Gaussian by $5.8\%$ and over KNN by $14.2\%$. These results indicate that explicitly encoding local linear geometry produces more semantically consistent neighborhoods, enhances robustness under noisy conditions, and mitigates spurious cross-class edges.

Beyond conventional baselines, we further compare against recent learnable and diffusion-based graph construction techniques, including GDC and IDGL. Although these methods incorporate global diffusion or end-to-end graph optimization, our model still attains higher accuracy across all datasets (e.g., by $1.4\%$ on WISDM and up to $1.0\%$ on HAR), suggesting that enforcing an affine manifold prior complements representation learning beyond generic structure learning or diffusion-based smoothing. In the following, we analyze graph-quality metrics (ICER, CCWS) and spectral properties to understand how geometry-aware construction improves message passing and depth stability.

Table 2: Graph quality metrics on HAR, HAPT, and WISDM. We report both structural purity (ICER, CCWS) and spectral statistics (low-/high-frequency ratios, spectral range). **Ours = VAE+LLE+GCN.** Lower ICER/CCWS and more balanced spectra are better.

| Dataset | Graph | Type | ICER $\downarrow$ | CCWS $\downarrow$ | LowFreq ($< 0.5$) | HighFreq ($> 1.5$) | Range |
|---------|-------|------|-------|-------|----------|-----------|-------|
| HAR | KNN | Static | 0.34 | 0.216 | 0.72 | 0.03 | 1.73 |
| | Cosine | Static | 0.44 | 0.353 | 0.74 | 0.07 | 1.70 |
| | A-KNN | Static | 0.29 | 0.124 | 0.61 | 0.19 | 1.78 |
| | Correlation | Static | 0.32 | 0.287 | 0.55 | 0.10 | 1.65 |
| | Gaussian | Static | 0.30 | 0.182 | 0.68 | 0.05 | 1.70 |
| | GDC | Diffusion | 0.28 | 0.122 | 0.46 | 0.20 | 1.79 |
| | IDGL | Dynamic | 0.22 | 0.108 | 0.43 | 0.23 | 1.80 |
| | **LLE (Ours)** | Dynamic | **0.21** | **0.093** | **0.42** | **0.28** | **2.00** |
| HAPT | KNN | Static | 0.33 | 0.205 | 0.75 | 0.02 | 1.65 |
| | Cosine | Static | 0.54 | 0.345 | 0.80 | 0.10 | 1.76 |
| | A-KNN | Static | 0.34 | 0.189 | 0.54 | 0.13 | 1.62 |
| | Correlation | Static | 0.58 | 0.413 | 0.68 | 0.09 | 1.74 |
| | Gaussian | Static | 0.29 | 0.176 | 0.70 | 0.04 | 1.65 |
| | GDC | Diffusion | 0.25 | 0.120 | 0.44 | 0.30 | 1.84 |
| | IDGL | Dynamic | 0.27 | 0.142 | 0.38 | 0.27 | 1.90 |
| | **LLE (Ours)** | Dynamic | **0.20** | **0.087** | **0.40** | **0.30** | **2.00** |
| WISDM | KNN | Static | 0.45 | 0.312 | 0.35 | 0.38 | 1.75 |
| | Cosine | Static | 0.66 | 0.421 | 0.38 | 0.26 | 1.70 |
| | A-KNN | Static | 0.52 | 0.375 | 0.29 | 0.40 | 1.80 |
| | Correlation | Static | 0.47 | 0.295 | 0.40 | 0.33 | 1.81 |
| | Gaussian | Static | 0.41 | 0.284 | 0.33 | 0.40 | 1.85 |
| | GDC | Diffusion | 0.30 | 0.140 | 0.30 | 0.42 | 1.88 |
| | IDGL | Dynamic | 0.37 | 0.180 | 0.35 | 0.43 | 1.83 |
| | **LLE (Ours)** | Dynamic | **0.29** | **0.148** | **0.28** | **0.45** | **2.00** |

## 4.2 GRAPH QUALITY EVALUATION

We evaluate whether LLE-based graphs yield cleaner, more class-consistent neighborhoods and more favorable spectral properties using the metrics defined in Section 3.3: (i) Inter-Class Edge Ratio (ICER, Eq. 7), (ii) Cross-Class Weight Share (CCWS, Eq. 10), (iii) spectral indicators (LowFreq, HighFreq, Range; Eqs. 11–12), and (iv) layer-wise cosine similarity (Eq. 13) to assess depth stability.

**Findings.** As shown in Table 2, our LLE-based graphs consistently outperform both static heuristics and recent learnable or diffusion-based approaches across all datasets. They achieve the lowest ICER and CCWS, indicating neighborhoods that are structurally purer (fewer inter-class edges) and more semantically concentrated (less cross-class mixing) than those produced by KNN, Gaussian, or Cosine graphs. Relative to GDC and IDGL, our method further reduces inter-class connectivity while maintaining a larger spectral range, reflecting a more balanced distribution of low- and high-frequency components. This spectral balance is strongly correlated with delayed over-smoothing and more stable information propagation. We next investigate how these structural and spectral properties influence the depth behavior of GNNs.

## 4.3 OVER-SMOOTHING ANALYSIS

We further examine over-smoothing by measuring the layer-wise cosine similarity of node embeddings across different graph construction strategies. Figures 2–4 show that static heuristics such as KNN, Gaussian, and Cosine lead to rapid feature homogenization, with similarity scores saturating after only 4–5 layers. Diffusion-based methods like GDC and learnable approaches such as IDGL alleviate this effect to some extent, but still exhibit relatively fast convergence as depth increases. In contrast, our LLE-based graphs homogenize significantly more slowly, preserving feature diversity over deeper layers and maintaining more discriminative representations throughout the network.

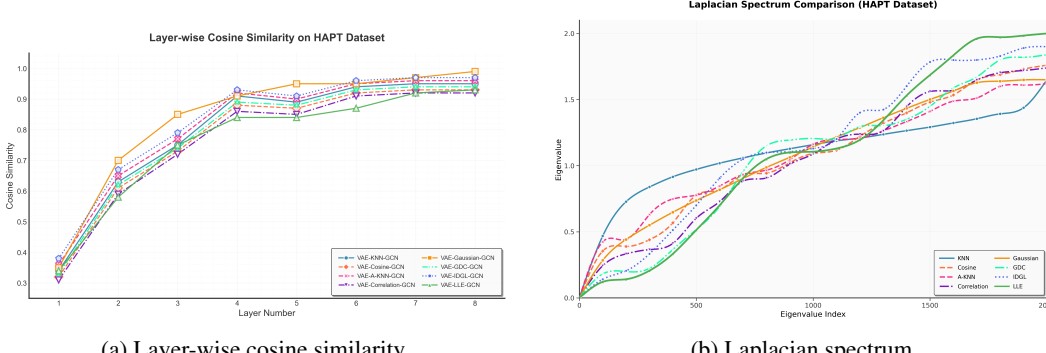

(a) Layer-wise cosine similarity.                    (b) Laplacian spectrum.

Figure 2: **HAPT oversmoothing analysis. (a)** Layer-wise cosine similarity: static heuristics (KNN, Gaussian, Cosine) exhibit rapid feature homogenization, with similarities saturating after 4–5 layers. Diffusion-based (GDC) and learnable (IDGL) methods alleviate this to some extent but still converge relatively quickly. In contrast, LLE-based graphs homogenize more slowly, preserving feature diversity even at greater depths. **(b)** Laplacian spectrum: LLE expands the spectral range ($\sim$2.0) and increases high-frequency components, producing a more balanced spectral energy distribution than all baselines. Together, these results demonstrate that geometry-aware graph construction mitigates over-smoothing by suppressing cross-class mixing and promoting more effective message propagation both structurally and spectrally.

This indicates that explicitly encoding local linear geometry mitigates excessive feature mixing and enables deeper GNNs to retain expressive power.

As shown in Figures 2b–4b, the Laplacian spectra of LLE-based graphs display a more balanced distribution of low- and high-frequency components and an expanded spectral range, suggesting that message passing is less dominated by low-frequency smoothing. Together, the spectral and layer-wise analyses provide strong empirical evidence that geometry-aware graph construction delays over-smoothing and enhances depth stability in deep GNNs.

## 5    CONCLUSION AND LIMITATIONS

**Conclusion.**    We proposed an end-to-end semi-supervised framework that integrates variational autoencoders, locally linear embedding (LLE), and graph neural networks. By enforcing local affine reconstruction during graph updates, our approach connects manifold geometry with message passing. Analytically, we showed that LLE attenuates spurious edges and balances spectral energy, thereby helping to mitigate over-smoothing. Empirically, on three wearable-sensor benchmarks (HAR, HAPT, WISDM), our method consistently improved accuracy and robustness, evidenced by lower cross-class contamination and more stable behavior at greater depth.

**Limitations.**    Despite these promising results, several limitations remain. First, our method requires $k$NN search in the latent space to establish neighborhood structure, which is a common bottleneck in graph construction and may become computationally expensive on large-scale datasets. Second, our evaluation focuses on wearable-sensor HAR datasets, which are naturally aligned with local affine modeling; generalization to other domains such as citation or molecular graphs may require further adaptation. Third, while the graph structure is periodically updated to reflect evolving representations, the update frequency remains coarse and does not yet capture fine-grained temporal dynamics, which may limit applicability to streaming or rapidly evolving graphs. Finally, although our spectral analysis provides evidence of alleviated over-smoothing, the precise trade-offs among variance, robustness, and depth stability remain an open question for future study.We believe our approach lays the groundwork for further advances in geometry-aware GNNs, and we discuss additional future directions in the Appendix  F.

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

APPENDIX

This appendix provides extended related work, analytical results, additional experiments, dataset details, and implementation notes to complement the main paper.

## A  EXTENDED RELATED WORK

### A.1  SEMI-SUPERVISED GNNS AND OVER-SMOOTHING (EXTENDED)

Graph Neural Networks (GNNs) such as the graph convolutional network (GCN) (Kipf & Welling, 2017) and attention-based variants like Graph Attention Networks (GAT) (Veličković et al., 2018) have demonstrated the effectiveness of message passing for semi-supervised node classification. However, a well-known limitation is that stacking many graph convolution layers leads to *over-smoothing*, where node representations become nearly indistinguishable across the graph (Zhao & Akoglu, 2020). Early theoretical results showed that as layers increase, GNN embeddings tend to converge to a constant subspace (Oono & Suzuki, 2021; Cai & Wang, 2020). More recent analyses have provided deeper insights: for example, a non-asymptotic study characterized precisely how and when over-smoothing occurs even at finite depths (Wu et al., 2023b), and further clarified the difference between over-smoothing and the related *over-squashing* phenomenon. Over-squashing refers to the bottleneck that arises when distant information is "squeezed" through limited graph pathways, impeding message propagation in deep GNNs (Alon & Yahav, 2021). These findings underscore the challenges of naively increasing GNN depth.

To combat over-smoothing, various architectural remedies have been proposed. Notable strategies include incorporating residual or skip connections to preserve information from earlier layers (Xu et al., 2018), using normalization techniques to maintain variance in node features across layers (Kelesis et al., 2025), and adopting personalized propagation schemes that mix initial features at every layer (Klicpera et al., 2019b). For instance, skip-connections or "jumping knowledge" networks allow a GNN to adaptively aggregate information from different layer depths (Xu et al., 2018), while PairNorm normalization was designed explicitly to prevent all node embeddings from collapsing to the same value by re-scaling layer outputs (Zhao & Akoglu, 2020). Personalized PageRank-based propagation (as in APPNP) also effectively mitigates over-smoothing by introducing a fixed teleport probability that retains each node's own features through the propagation process (Klicpera et al., 2019b). These techniques significantly improve the training of deeper GNNs. However, it is worth noting that they are general-purpose architectural fixes – they do not explicitly inject any geometric prior about the input space into the model. In other words, while such methods help avoid representation collapse, they do not leverage manifold structure in the data. This leaves an open question of whether encoding geometric information (e.g., local manifold structure of the features) can further alleviate issues like over-smoothing or enhance GNN performance, which is one of the motivations for our approach.

### A.2  GRAPH STRUCTURE LEARNING (EXTENDED)

Another line of work studies how to optimally learn or adjust the graph structure itself in tandem with GNN training. Instead of assuming a fixed input adjacency, these approaches aim to learn a better graph that can yield improved downstream predictions. Franceschi et al. (Franceschi et al., 2019) introduced a bi-level optimization framework for learning discrete graph structures in a differentiable manner. Their method (LDS) samples graph structures during training and backpropagates through a discrete edge distribution, thereby identifying useful connections. Since then, a variety of graph structure learning (GSL) approaches have been proposed to address settings where the initial graph is noisy, incomplete, or even unavailable (Zhu et al., 2022; Xia et al., 2021; Barbero et al., 2024).

Recent work has pushed this further in several respects. For example, SE-GSL (Zou et al., 2023) leverages structural entropy and hierarchical encoding trees to enhance robustness and interpretability, including on noisy or heterophilic graphs. Within the OpenGSL benchmark suite, OpenGSL (Zhou et al., 2023b) compresses mutual information to learn a compact graph and reports robust semi-supervised performance under attacked/noisy settings. In another direction, a geometry-enhanced GNN for glassy dynamics (Jiang et al., 2023) augments message passing with rotation-invariant distance and angular features to encode smoothness patterns and geometric rela-

tions. Collectively, these studies suggest the benefits of combining geometry-aware criteria with structure learning.

Despite these advances, most such approaches still rely on implicit or heuristic priors (e.g., entropy, mutual information, feature similarity) rather than explicitly modeling geometric manifold structure (affine/local linear patches) during graph construction. Additionally, many assume the graph structure is static (learned once or preprocessed) rather than being dynamically updated during GNN training. Some recent works begin to address this gap—for example, DBGSL for end-to-end dynamic brain graph structure learning (Campbell et al., 2024), DG-Mamba for robust dynamic graph learning in spatio-temporal settings (Yuan et al., 2025), and diffusion-based manifold alignment methods (Rhodes & Rustad, 2024). This leaves room for approaches that integrate explicit geometric priors with dynamic graph updates in an end-to-end framework, aligning directly with the motivations of our work.

### A.3 MANIFOLD METHODS AND LLE (EXTENDED)

Our work is inspired by the literature on manifold learning and geometry-preserving dimensionality reduction, which has developed somewhat separately from graph neural networks. Classical manifold learning techniques such as Locally Linear Embedding (LLE) (Roweis & Saul, 2000; Saul & Roweis, 2003) and Laplacian Eigenmaps (Belkin & Niyogi, 2003) aim to obtain low-dimensional embeddings of high-dimensional data while preserving local neighborhood structure. LLE, for example, assumes that each data point can be approximated by a linear combination of its nearest neighbors; it finds coordinates that minimize this reconstruction error, producing an embedding that retains local linear relationships. The affinity weights derived in LLE can be interpreted as defining a sparse graph that reflects intrinsic neighborhood relations on the underlying manifold. Laplacian Eigenmaps similarly construct a nearest-neighbor graph and then compute a spectral embedding that preserves local pairwise distances. These methods yield graphs or weight matrices grounded in the geometry of the input data (e.g., capturing non-linear manifold structure) and are widely used and empirically effective for unsupervised learning and visualization. However, they are typically applied as separate preprocessing steps rather than being integrated into modern end-to-end deep learning pipelines. A few initial attempts have sought to combine manifold embedding ideas with deep networks—for instance, leveraging Hessian LLE features within a GNN for image analysis (Jayapal & Annamalai, 2025)—but such examples remain uncommon. By and large, explicit manifold-based priors are still rarely enforced in mainstream GNN frameworks for graph construction or message passing.

## B EXTENDED ANALYTICAL RESULTS

### B.1 LOCAL AFFINE RECONSTRUCTION AND CROSS-CLASS MIXING

LLE computes weights by minimizing a local reconstruction residual subject to an affine constraint. This favors neighbors that reside on the same local manifold patch; cross-class neighbors typically introduce a normal component to the local tangent space and therefore receive smaller weights. Two quantities are naturally associated with this process: (i) the average LLE reconstruction residual $\frac{1}{n}\sum_i \|z_i - \sum_{j\in\mathcal{N}(i)} w_{ij}z_j\|_2^2$, and (ii) the cross-class weight share $\sum_i \sum_{j:y_j\neq y_i} w_{ij}$. In principle, these quantities are expected to decrease together: reducing reconstruction residual implies that most weight is assigned to neighbors lying on the same manifold patch, which in turn reduces cross-class mixing.

### B.2 SPECTRAL ENERGY BALANCE AND DEPTH STABILITY

Graph propagation contracts feature components according to the Laplacian spectrum. Heuristic graphs (e.g., $k$NN/Gaussian) often concentrate energy in low-frequency modes, accelerating homogenization. The LLE-induced graph tends to expand the spectral range and reallocate mass to higher frequencies, which slows the collapse of non-trivial components, thereby delaying over-smoothing. We corroborate this with spectral statistics in the main text and extended plots in Appendix C.

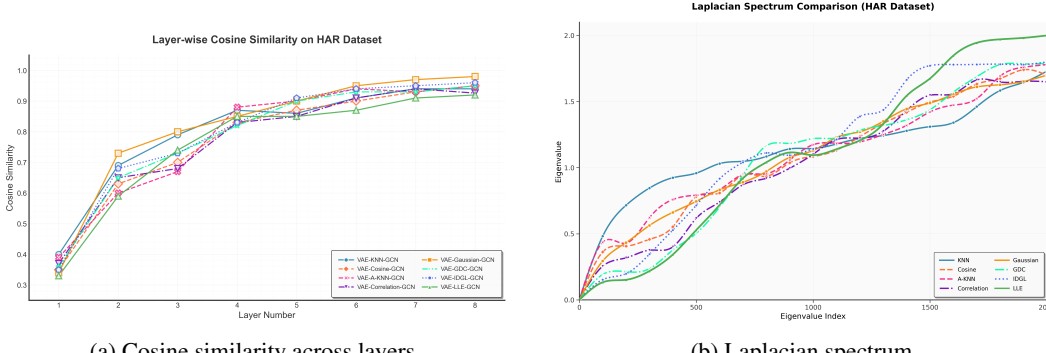

(a) Cosine similarity across layers.

(b) Laplacian spectrum.

Figure 3: **HAR oversmoothing analysis. (a)** Layer-wise cosine similarity: static heuristics (KNN, Gaussian, Cosine) exhibit rapid feature homogenization, with similarities saturating after 4–5 layers. Diffusion-based (GDC) and learnable (IDGL) methods slow down this process but still converge earlier than our LLE-based approach, which preserves feature diversity even at greater depths. **(b)** Laplacian spectrum: LLE expands the spectral range ($\sim$2.0) and increases high-frequency components, leading to a more balanced spectral energy distribution across frequencies. Together, these results indicate that geometry-aware graph construction mitigates over-smoothing and sustains discriminative capacity in deeper GNNs.

### B.3 SCOPE

We intentionally avoid claiming new general-purpose theorems; instead, we connect known geometric/spectral intuitions with quantities our method directly controls and reports in experiments (cross-class mixing, spectra, layer-wise similarity).

## C ADDITIONAL EXPERIMENTAL RESULTS

We provide supplementary results to complement those reported in Section 4 of the main text.

### C.1 LAYER-WISE COSINE SIMILARITY AND SPECTRUM

Figures 3 and 4 extend the HAPT analysis to the HAR and WISDM datasets, providing a more comprehensive view of over-smoothing behavior across benchmarks. Across all settings, LLE-based graphs consistently exhibit slower growth in layer-wise cosine similarity compared to static heuristics (KNN, Gaussian, Cosine) and even advanced approaches such as GDC and IDGL, indicating that feature diversity is preserved deeper into the network. Moreover, the Laplacian spectra show that LLE expands the spectral range and balances low- and high-frequency components more effectively, reflecting improved propagation dynamics. These results further support our conclusion that explicitly encoding local linear geometry mitigates over-smoothing and enhances depth stability in GNNs.

### C.2 NOISE ROBUSTNESS

Our framework is expected to improve robustness against input perturbations, since LLE-based graphs suppress spurious cross-class connections and emphasize geometry-consistent neighborhoods. This effect is most evident on the WISDM dataset, which is known to contain more noise due to its smartphone-based collection in unconstrained real-world settings. Compared to HAR and HAPT, where sensor placement is more controlled, WISDM poses stronger challenges with higher intra-class variability and cross-class contamination. As shown in Table 1, our dynamic LLE-based approach achieves a much larger accuracy gain on WISDM ($+6\%$ over Gaussian and $+14\%$ over KNN), confirming that geometry-aware graph construction leads to significantly improved robustness under noisy conditions.

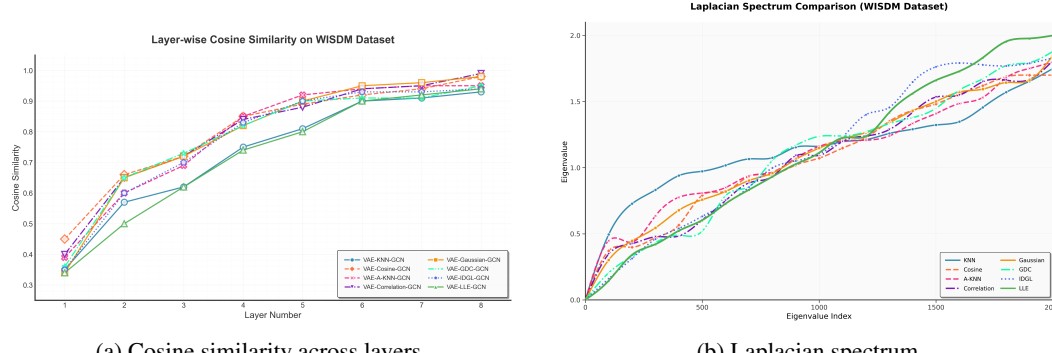

(a) Cosine similarity across layers.      (b) Laplacian spectrum.

Figure 4: **WISDM oversmoothing analysis. (a)** Layer-wise cosine similarity: traditional baselines (KNN, Gaussian, Cosine) homogenize rapidly, and while GDC and IDGL slow down convergence, embeddings still lose diversity as depth increases. In contrast, LLE-based graphs homogenize more gradually, maintaining discriminative feature representations even at 8 layers. **(b)** Laplacian spectrum: LLE widens the spectral range and achieves a more balanced low-/high-frequency distribution, indicating improved propagation dynamics and more stable message passing. These results demonstrate that incorporating local linear geometry significantly delays over-smoothing and enhances robustness in noisy real-world settings.

Table 3: **Comparison of HAR, HAPT, and WISDM.** Columns list the number of classes, features per window, windowing protocol, and preprocessing used in our pipeline. Unless noted otherwise, we follow the official subject-wise 70%/30% train/test split and carve out 10% of the training set for validation. For semi-supervised WISDM, we mask 80% of training labels and uniformly sample 15,000 unlabeled instances from the training partition (no overlap with validation/test). HAR/HAPT retain the released Butterworth filtering; we subsequently apply z-score to the 561-D features. WISDM uses 54 statistical features with z-score.

| Dataset | Classes | Features | Window | Preprocessing | Notes |
|---------|---------|----------|--------|---------------|-------|
| HAR | 6 | 561 | 2.56 s, 50% overlap | Butterworth; z-score | Lab setting |
| HAPT | 12 | 561 | 2.56 s, 50% overlap | Butterworth; z-score | Adds transitions |
| WISDM | 6 | 54 | statistical windowing | z-score | Real-world noise |

# D DATASET DETAILS

We summarize the datasets and our preprocessing for reproducibility.

## D.1 HAR (HUMAN ACTIVITY RECOGNITION USING SMARTPHONES)

The HAR dataset (Reyes-Ortiz et al., 2013) contains recordings from 30 volunteers performing six daily activities (*walking, walking upstairs, walking downstairs, sitting, standing, laying*). Each subject wore a waist-mounted smartphone with a tri-axial accelerometer and gyroscope. Signals were sampled at 50 Hz and segmented into fixed-length windows of 2.56 s (128 readings) with 50% overlap. A Butterworth filter was applied to separate body motion and gravity components, and 561 time- and frequency-domain features were extracted per window. The dataset comprises 10,299 instances. Following the official protocol, we use the 70%/30% train/test split and carve out 10% of the training data for validation.

## D.2 HAPT (HUMAN ACTIVITIES AND POSTURAL TRANSITIONS)

The HAPT dataset (Reyes-Ortiz et al., 2015) extends HAR by including 12 classes (six basic activities and six postural transitions) under the same acquisition protocol, yielding 561 features per window. It contains 10,929 instances. Following HAR, we use the official 70%/30% train/test split and carve out 10% of the training set for validation.

Table 4: Node classification accuracy (%) with the SimpleGNN backbone. Means $\pm$ std over multiple runs; best in bold.

| Method | HAR | HAPT | WISDM |
|---|---|---|---|
| *Static heuristics* | | | |
| VAE + KNN + SimpleGNN (static) | $80.2 \pm 0.86$ | $81.2 \pm 0.21$ | $64.0 \pm 0.42$ |
| VAE + Cosine + SimpleGNN (static) | $75.8 \pm 0.40$ | $78.4 \pm 0.36$ | $61.2 \pm 0.28$ |
| VAE + A-KNN + SimpleGNN (static) | $85.3 \pm 0.35$ | $87.5 \pm 0.24$ | $70.8 \pm 0.26$ |
| VAE + Correlation + SimpleGNN (static) | $82.1 \pm 0.33$ | $83.2 \pm 0.27$ | $66.4 \pm 0.30$ |
| VAE + Gaussian + SimpleGNN (static) | $86.4 \pm 0.22$ | $88.6 \pm 0.02$ | $72.3 \pm 0.31$ |
| *Learnable / diffusion-based graphs* | | | |
| VAE + GDC + SimpleGNN (diffusion) | $88.1 \pm 0.31$ | $89.3 \pm 0.45$ | $75.5 \pm 0.21$ |
| VAE + IDGL + SimpleGNN (dynamic) | $89.5 \pm 0.28$ | $89.6 \pm 0.22$ | $74.1 \pm 0.19$ |
| **VAE + LLE + SimpleGNN (dynamic)** | $\mathbf{89.9 \pm 0.12}$ | $\mathbf{90.8 \pm 0.20}$ | $\mathbf{77.2 \pm 0.12}$ |

Table 5: **Sensitivity analysis of LLE hyperparameters across datasets.** Classification accuracy (%) with varying neighborhood size $k \in \{10, 20, 30, 40\}$ and regularization coefficient $\epsilon \in \{0.1, 0.01, 0.001\}$. Results are reported as mean $\pm$ std over multiple runs. Bold indicates the best setting per dataset.

| Dataset | $\epsilon$ | $k = 10$ | $k = 20$ | $k = 30$ | $k = 40$ |
|---|---|---|---|---|---|
| HAR | 0.1 | $89.8 \pm 0.30$ | $\mathbf{95.2 \pm 0.23}$ | $93.5 \pm 0.32$ | $91.0 \pm 0.25$ |
| | 0.01 | $89.2 \pm 0.29$ | $91.7 \pm 0.17$ | $92.1 \pm 0.40$ | $90.5 \pm 0.30$ |
| | 0.001 | $88.4 \pm 0.22$ | $90.2 \pm 0.30$ | $92.0 \pm 0.21$ | $89.9 \pm 0.28$ |
| HAPT | 0.1 | $90.9 \pm 0.24$ | $\mathbf{94.3 \pm 0.02}$ | $92.7 \pm 0.11$ | $90.1 \pm 0.36$ |
| | 0.01 | $86.9 \pm 0.45$ | $90.2 \pm 0.28$ | $91.2 \pm 0.05$ | $89.5 \pm 0.45$ |
| | 0.001 | $86.1 \pm 0.12$ | $89.4 \pm 0.52$ | $90.7 \pm 0.51$ | $88.3 \pm 0.32$ |
| WISDM | 0.1 | $81.6 \pm 0.17$ | $81.2 \pm 0.12$ | $\mathbf{82.2 \pm 0.04}$ | $79.9 \pm 0.29$ |
| | 0.01 | $79.4 \pm 0.18$ | $80.7 \pm 0.26$ | $81.0 \pm 0.19$ | $80.4 \pm 0.11$ |
| | 0.001 | $80.2 \pm 0.55$ | $80.9 \pm 0.29$ | $80.2 \pm 0.30$ | $79.5 \pm 0.12$ |

### D.3 WISDM (WIRELESS SENSOR DATA MINING)

The WISDM dataset (Weiss, 2019) provides accelerometer recordings collected in unconstrained real-world settings with six activity classes. Raw traces are segmented into windows and transformed into 54 statistical features. The labeled portion contains 5,435 samples, and the unlabeled pool has approximately 1.37M samples. For semi-supervised experiments, we follow the same protocol as HAR: we use the official 70%/30% train/test split and carve out 10% of the training set for validation. From the unlabeled pool (restricted to the training partition and the provided unlabeled set with no overlap to validation/test), we uniformly sample 15,000 instances *without replacement* using a fixed random seed, and release the indices for reproducibility.

**Preprocessing (all datasets).** Unless otherwise noted, we re-standardize all features with z-score before model training to ensure comparability across datasets (see Table 3 for acquisition, windowing, and dataset-specific preprocessing).Normalization statistics are fitted on the *training* partition (including unlabeled samples when applicable) and then applied to validation/test to avoid leakage.

## E IMPLEMENTATION DETAILS

### E.1 TRAINING SETUP

We implement all models in PyTorch Geometric. The VAE is optimized with Adam ($\mathrm{lr} = 3 \times 10^{-3}$, weight decay $= 1 \times 10^{-4}$). The GNN is trained in *full-batch* on the current graph. Early stopping

uses a patience of 100 epochs based on validation accuracy. Unless otherwise noted, the latent dimension is 128 and activations are ReLU. At the beginning of every epoch, we recompute LLE weights on the current latent codes and rebuild the sparse adjacency (dynamic update). Note that the LLE weight matrix $W$ and the normalized adjacency $\hat{A}$ are recomputed in closed form at each epoch and are not updated by backpropagation; only the encoder, decoder, and GNN parameters receive gradient updates. For the joint objective, we set $\lambda_{\mathrm{VAE}} = \lambda_{\mathrm{LLE}} = 0.2$. For reproducibility, code will be released upon paper acceptance.

### E.2 Additional Results with SimpleGNN

For completeness, we also evaluate a lightweight SimpleGNN backbone under the same dynamic graph-construction protocol (LLE recomputed every epoch). As shown in Table 4, LLE-based dynamic graphs consistently outperform static KNN and Gaussian baselines, indicating that the benefit of geometry-aware graphs is not tied to a particular GNN backbone. This further suggests that our framework is architecture-agnostic and can generalize to lightweight or resource-constrained settings, where SimpleGNN may be preferable.

**Parameter sensitivity.** We analyze the effect of two key hyperparameters in LLE-based graph construction. Specifically, we vary the neighborhood size $k \in \{10, 20, 30, 40\}$ and the LLE regularization coefficient $\epsilon \in \{0.1, 0.01, 0.001\}$. Table 5 summarizes the results (means $\pm$ std over multiple runs). Overall, performance is relatively stable across a broad range: HAR and HAPT peak at $k = 20$ with $\epsilon = 0.1$, while WISDM attains its best at $k = 30$ with $\epsilon = 0.1$.

## F Future Work and Outlook

Our framework opens up several promising research directions beyond the current scope. One is to scale LLE-based graph construction to million-node scenarios, improving computational efficiency and enabling deployment on large-scale real-world graphs. Another is to extend the framework to real-time temporal activity recognition, where neighborhood structures evolve continuously. Finally, integrating affine reconstruction with attention-based propagation could bridge local geometric priors with adaptive long-range message passing, potentially further improving representational power.

