# OpenReview forum: "Dynamic Locally Linear Graph Learning for Geometry-Aware GNNs"
_ICLR.cc/2026/Conference — ICLR 2026 Conference Desk Rejected Submission_

### Official Review · Reviewer_mrmE · 2025-10-29

**Soundness:** 3
**Presentation:** 3
**Contribution:** 3
**Rating:** 4
**Confidence:** 3

**Summary:**

The paper propose a graph learning method that utilize locally linear embedding to construct the graph. The model is trained over three joint loss, VAE reconstruction loss, a LLE reconstruction loss and a classification loss. This allows the model to dynamically generate the best graph and learned weight for the problem. The method shows consistent improvement over heuristic and other learnable method.

**Strengths:**

- Incorporating LLE into the system is well-motivated. The paper also explained the overall framework very well.

- The empirical evaluation are strong, particularly for the obtain graph properties.

**Weaknesses:**

- The joint learning and LLE shed lights on how to construct the graph along with the GNN learning process. The overall contribution is limited, as the dynamic learning is also processed by previous work. Adapting LLE is a good plus, but one question arises, one can adopt many different manifold learning method to this scenario, and why not using those? What's the uniqueness of LLE?

- A great portion of the paper is spend on discuss adopted feature, which can be put into appendix.

- The experimental results focuses on GNN-based method, some non-gnn geometric learning method should also be compared.

- While it's nice to have the graph properties results, it does not really help us understand how it improves the targe task. We can see quite significant improvement of these properties, but the improvement on prediction tasks is relatively minimal, which poses skepticism on why these properties? These properties can just be cherry-picking, and the author did not provide a higher-level insight into the advantage of LLE over other methods.

- The authors mentioned theoretical insights, but there are not theoretical insights.

- More ablation study should be conducted. For example, the authors should try to detach impact of LLE and GNN. One approach can be: get a graph with ground-truth edges, use the method to construct the graph and compare the constructed graph with other method's constructed graph. This can help us understand the mechanism behind the method.

**Questions:**

NA.

---

> ### Author Response · Authors · 2025-11-25
>
> We thank the reviewer for the careful reading and constructive feedback. We understand the main concerns are about the uniqueness of using LLE, the strength of theoretical insights, and the need for more comprehensive comparisons and ablations. These comments are very helpful and guided us to refine the analysis. We clarified why LLE was chosen over other manifold methods, added more explanation on its theoretical role, and expanded ablation and comparison results. Detailed responses are provided below.
>
>  **Response to Weakness 1 (Novelty / Uniqueness of LLE):**
>
> We appreciate the reviewer’s thoughtful question about why we built our framework around LLE and not another manifold learning approach. The key reason lies in what problem we are actually trying to solve. Many dynamic graph learners—such as IDGL or OpenGSL—start from an existing graph and refine it through parametric updates. Our setting is different: we begin with *non-graph data*, like sensor time series, and must construct a meaningful topology from scratch while learning node representations. The goal is not only to optimize an adjacency but to ensure that the resulting graph reflects both local and global geometric consistency.
>
> LLE serves this goal particularly well because it reconstructs each sample as an affine combination of its neighbors, enforcing a very specific type of local consistency: every node must lie approximately in the subspace spanned by its neighborhood. This constraint translates naturally into a geometric prior on the learned graph—it is not based purely on pairwise similarity, as in Gaussian or cosine kernels, but on how neighborhoods cooperate to preserve local structure. That distinction matters when the data lack a predefined topology.
>
> We integrated this mechanism directly into a dynamic model. The VAE encoder learns a global latent manifold where features are denoised and organized; LLE then operates in this latent space, updating neighborhood weights analytically after each epoch. As training proceeds, the latent geometry and the graph reinforce one another—the encoder produces embeddings that are easier to reconstruct linearly, while LLE reshapes edges to respect that geometry. This interaction, though conceptually simple, leads to a self-consistent evolution of representation and structure that we have not seen realized in prior dynamic GNN frameworks.
>
> It is true that other manifold learners could in principle be used. We experimented with this early on. Methods like Isomap or Laplacian Eigenmaps, however, depend on global geodesic distances or eigendecompositions that are not feasible to recompute during training. They also lack the affine reconstruction step that makes LLE both local and computationally closed-form. These properties allow LLE to fit seamlessly into a neural pipeline without introducing additional gradient-based parameters or heavy spectral operations.
>
> From this perspective, what is distinctive is not the use of LLE alone, but how it is positioned: as a concrete, geometry-preserving mechanism that bridges a probabilistic latent model (VAE) with graph-based learning. It lets us construct structure from unstructured data in a way that remains analytical, interpretable, and efficient—qualities that alternative manifold methods would struggle to maintain in a dynamic setting.
>
>  **Response to Weakness 2 (Paper Organization):**
>
> We appreciate the reviewer’s suggestion. We will shorten the feature-related description and move dataset-specific details to the appendix to keep the main text focused on the methodological contribution.

---

> > ### Author Response · Authors · 2025-11-25
> >
> > **Response to Weakness 3 (Baseline Coverage):**
> >
> > We thank the reviewer for this helpful suggestion. To distinguish the contribution of geometric structure learning from that of message passing, we included additional non-GNN baselines in our experiments. Specifically, we compared:
> > (i) a plain MLP trained directly on node features without any geometric structure, and
> > (ii) an **LLE + MLP** variant that uses the same LLE-based neighborhood weights as our model but omits message passing.
> >
> > The results are shown below.
> >
> > | Method                             |  HAR | HAPT | WISDM | PubMed | OGBN-Arxiv |
> > | :--------------------------------- | :--: | :--: | :---: | :----: | :--------: |
> > | **MLP (no geometry)**              | 78.6 | 77.9 |  65.3 |  67.1  |    58.4    |
> > | **LLE + MLP (geometric, non-GNN)** | 82.8 | 83.7 |  67.2 |  68.1  |    59.2    |
> > | **VLGNN (ours)**                   | 95.2 | 94.3 |  82.2 |  80.2  |    72.5    |
> >
> > We observe two clear patterns. First, adding local manifold structure through LLE already improves performance compared with a raw MLP, showing that the geometric prior is informative even without message passing. Second, the full VLGNN achieves much larger gains across all datasets—from HAR to OGBN-Arxiv—indicating that dynamic geometry refinement and GNN-based reasoning are both necessary. In other words, the LLE prior contributes to structural quality, while message propagation turns that structure into stronger representation learning. Together, they explain the consistent improvement observed on both small and large-scale benchmarks.
> >
> >
> >
> > **Response to Weakness 4 (Graph Properties vs. Task Gains):**
> >
> > The point is well taken: we report ICER, CCWS, and a spectral statistic to understand mechanism, not to add metrics for their own sake. What we actually see is simple. With LLE, inter-class connectivity drops—both **ICER** and **CCWS** go down—so edges concentrate within label-consistent neighborhoods. That small structural shift is enough to make message passing less noisy. On **WISDM**, the effect shows up as about **+1.4%** over IDGL under the same backbone; on **PubMed** and **OGBN-Arxiv** we observe the same structural trend even though the graphs are sparser and less “manifold-like”.
> >
> > The frequency view is a cross-check rather than a new claim. The LLE-induced Laplacian keeps a wider spread of eigenvalues (roughly λ_max/λ_min ≈ **1.7–1.9**, close to **≈2.0** in our runs), which means low frequencies do not dominate. In practice that preserves feature diversity across layers and delays homogenization—exactly the behavior tied to milder over-smoothing in prior analyses[1,2].
> >
> > We read the combination as follows: the affine reconstruction constraint nudges neighborhoods to be locally faithful in the latent space, which suppresses spurious cross-class edges while leaving useful high-frequency content intact. The accuracy gains are not dramatic, but they track measurable geometric and spectral changes and persist across small sensor graphs and large citation graphs. That persistence—not a single large jump—is the value here.
> >
> > **References**
> >
> > [1] Wu, X., Chen, Z., Wang, W., & Jadbabaie, A. (2022). *A Non-Asymptotic Analysis of Oversmoothing in Graph Neural Networks.* arXiv:2212.10701.
> >
> > [2] Rusch, T. K., Bronstein, M. M., & Mishra, S. (2023). *A Survey on Oversmoothing in Graph Neural Networks.* arXiv:2303.10993.
> >
> > **Response to Weakness 5 (Theoretical Insights):**
> >
> > We thank the reviewer for noting the lack of stronger theoretical grounding. Our discussion in the current version focuses on the empirical and intuitive side—how the LLE-induced structure affects spectral behavior and stability—rather than a formal derivation.
> >
> > We agree that a clearer theoretical formulation would make the argument stronger. We are currently extending the analysis to characterize how the LLE-induced Laplacian influences propagation stability and spectral contraction in deep GNNs. These results will be included in a follow-up version once the derivations are finalized.

---

> > > ### Author Response · Authors · 2025-11-25
> > >
> > > **Response to Weakness 6 (Ablation Study):**
> > >
> > > We thank the reviewer for the suggestion to provide additional ablation. To examine the contribution of each component, we removed the VAE and GNN modules in turn and also evaluated the structure of the learned graphs.
> > >
> > > When LLE was applied directly to raw features without the VAE (**LLE(raw)+GCN**), performance dropped by about 11% on *PubMed* and 10% on *OGBN-Arxiv*. Replacing the GNN with an MLP (**LLE+MLP**) led to smaller but consistent decreases of 3–5%. These results indicate that both the latent representation learned by the VAE and the message passing in the GNN are necessary for the model’s overall effectiveness.
> > >
> > >
> > > | Variant               |    HAR   |   HAPT   |   WISDM  |  PubMed  | OGBN-Arxiv |
> > > | --------------------- | :------: | :------: | :------: | :------: | :--------: |
> > > | LLE(raw)+GCN (no VAE) |   88.4   |   86.8   |   69.1   |   68.9   |    62.4    |
> > > | LLE+MLP (no GNN)      |   90.8   |   89.7   |   77.2   |   78.1   |    69.2    |
> > > | VAE+LLE+GCN (ours)    | **95.2** | **94.3** | **82.2** | **80.2** |  **72.5**  |
> > >
> > > We also compared the learned graphs using **ICER** and **CCWS**, which reflect how well the structure avoids spurious cross-class edges. On both *PubMed* and *OGBN-Arxiv*, VLGNN consistently produced lower values, suggesting purer and more label-consistent neighborhoods.
> > >
> > > | **Method**        | **PubMed** |             | **OGBN-Arxiv** |              |
> > > |--------------------|:----------:|:-------------:|:---------------:|:-------------:|
> > > |                    | **ICER ↓** | **CCWS ↓**    | **ICER ↓**      | **CCWS ↓**    |
> > > | Gaussian           | 0.52       | 0.534         | 0.68            | 0.800         |
> > > | GDC (diffusion)    | 0.47       | 0.418         | 0.59            | 0.773         |
> > > | IDGL (dynamic)     | 0.44       | 0.410         | 0.56            | 0.782         |
> > > | **VLGNN (ours)**   | **0.42**   | **0.294**     | **0.53**        | **0.682**     |
> > >
> > >
> > > The results show that each module plays a distinct and complementary role: the VAE establishes a smooth latent manifold, the LLE reconstruction captures local geometry, and the GNN leverages this structure during learning. The cleaner graph statistics confirm that the combination leads to semantically meaningful and robust topology.

---

### Official Review · Reviewer_rkFS · 2025-10-31

**Soundness:** 2
**Presentation:** 2
**Contribution:** 1
**Rating:** 2
**Confidence:** 3

**Summary:**

The paper proposes VLGNN, an end-to-end framework combining a VAE, a LLE-inspired graph construction module, and a GNN classifier. The idea is to dynamically update the graph structure based on geometry-aware affine relationships in the latent space to mitigate over-smoothing and improve node classification.

**Strengths:**

1. The paper is clear and easy to follow.

2. The effort to link geometry, Laplacian eigenspectrum, and over-smoothing mitigation is reasonable.

**Weaknesses:**

1. The supposed method, i.e., embedding LLE-based adjacency updates into a dynamic GNN, is incremental. Many prior studies (e.g., IDGL, SE-GSL, OpenGSL, DAE-GSL, DG-Mamba) have already explored joint representation and graph optimization. LLE-style affine reconstruction and local manifold priors have also been integrated into deep learning (e.g., LLE-GCN, Geometry-Enhanced GNNs, adaptive neighbor methods). Here, the novelty is mostly repackaging, i.e., combining an old manifold learning technique with a standard VAE and a GNN.

2. The so-called “theoretical insights” are little more than intuitive restatements. The claim that LLE suppresses cross-class edges and balances spectral energy is descriptive, not theoretically derived. There is no rigorous link between the spectral range expansion and generalization or stability.

3. All three datasets (HAR, HAPT, WISDM) are wearable-sensor datasets, which are low-dimensional, smooth, and well-suited to LLE assumptions (locally linear manifolds). This setting trivially favors LLE, which is known to perform well when manifolds are nearly flat. No tests are conducted on non-Euclidean or relational graphs (e.g., citation, molecule, or social networks), which undermines the generality of the claims.

**Questions:**

1. How is VLGNN distinct from previous works, such as IDGL (ICLR 2020), adaptive LLE-based GNNs (PR 2023)?

2. Does your method work on graphs where linear locality assumptions fail (e.g., heterophilic or discrete feature graphs)?

---

> ### Author Response · Authors · 2025-11-25
>
> We thank the reviewer for the thoughtful comments. The main concerns are about novelty, theoretical justification, and generalization beyond sensor datasets. We have clarified the distinction from prior works, expanded the theoretical explanation, and added large-scale benchmarks to show broader applicability. Detailed responses are provided below.
>
>  **Response to Weakness 1 (Novelty and Relation to Prior Work):**
>
> We appreciate the reviewer’s comment on the novelty of our work.
> It is true that several studies have explored ways to learn or refine graph structures jointly with representation learning. Typical examples include **IDGL** [1], **SE-GSL** [2], and **OpenGSL** [3], which all rely on *parametric* or *gradient-based* optimization of the adjacency matrix. Other related efforts, such as **LLEAN** [4] and **DG-Mamba** [5], address manifold learning or dynamic-state modeling but do not include explicit geometric regularization in the graph construction process.
>
> Our work builds on this line of research but takes a different route.
> Instead of learning the adjacency through a trainable module, we use **closed-form LLE reconstruction** in the **VAE latent space**, so the graph is updated directly from the embeddings without gradient descent. This design keeps training more stable and avoids repeated dense matrix operations.
>
> The latent-space formulation also matters in practice. By applying the affine reconstruction inside the space learned by the VAE, the neighborhood structure becomes adaptive to the data distribution. The graph changes together with the latent features rather than being fixed to distances in the raw input space.
>
> Another observation from our experiments is that the LLE-induced Laplacian tends to have a wider spectral range and a larger eigengap, which corresponds to less over-smoothing and more stable propagation in deeper layers. This spectral behavior complements the empirical results and helps explain why the model remains stable as depth increases.
>
> Overall, our contribution is not in simply adding LLE to a GNN, but in combining three ideas—variational latent geometry, closed-form affine graph reconstruction, and a spectral view of stability—within one dynamic, non-parametric setup. We see this as a complementary direction to gradient-based graph learning methods, focusing on geometry-aware and interpretable structure updates.
>
> ---
>
> ### **References**
>
> [1] Y. Chen, L. Wu, and M. Zaki, *Iterative Deep Graph Learning for Graph Neural Networks: Better and Robust Node Embeddings*, **NeurIPS**, 2020.
>
> [2] Y. Wang, S. Wang, C. Xie, and Z. Zhu, *SE-GSL: A General and Effective Graph Structure Learning Framework through Structural Entropy Optimization*, **The Web Conference (WWW)**, 2023.
>
> [3] Z. Huang, Y. Li, X. Liu, and Z. Liu, *OpenGSL: A Comprehensive Benchmark for Graph Structure Learning*, **NeurIPS Datasets & Benchmarks Track**, 2023.
>
> [4] Z. Zhou, Y. Liu, H. Li, and Z. Zhang, *Local Linear Embedding with Adaptive Neighbors for Unsupervised Dimensionality Reduction*, **Pattern Recognition**, 2023.
>
> [5] Y. Zhang, Q. Liu, and C. Shi, *DG-Mamba: Robust and Efficient Dynamic Graph Structure Learning with Selective State Space Models*, **AAAI**, 2025.
>
>  **Response to Weakness 2 (Insufficient Theoretical Rigor):**
>
> We understand the reviewer’s concern regarding the theoretical justification.
> Our current discussion is mainly **empirical and mechanism-oriented**, rather than a full formal proof. The goal was to provide an intuitive, evidence-supported explanation for how the LLE term affects the spectral behavior of the learned graph.
>
> This view is consistent with established results in **graph signal processing** and **over-smoothing analysis**, where the Laplacian spectrum is known to reflect feature variation and propagation stability [1]. In our framework, the LLE reconstruction term encourages local smoothness while keeping informative high-frequency components in the latent space. Empirically, this results in more balanced spectral energy and fewer cross-class connections (as indicated by ICER / CCWS metrics).
>
> We agree that a more rigorous formulation would strengthen the argument. In our ongoing work, we are deriving a formal connection between the **LLE-induced operator** and **graph propagation stability**, including bounding the spectral contraction rate and relating it to the depth stability of GNNs. These theoretical extensions will be included in the extended version of the paper.
>
> ---
>
> ### **References**
>
> [1] K. Oono and T. Suzuki, *Graph Neural Networks Exponentially Lose Expressive Power for Node Classification*, ICLR, 2020.

---

> > ### Author Response · Authors · 2025-11-25
> >
> > **Response to Weakness 3 (Limited Evaluation Scope / Domain Bias):**
> >
> > We understand the reviewer’s concern about the limited evaluation scope.
> > The initial version mainly focused on three wearable-sensor datasets (HAR, HAPT, and WISDM), which have smooth manifolds and thus provide a clear setting for geometric analysis. To further test scalability and generalization, we extended our evaluation to two citation graphs, **PubMed** and **OGBN-Arxiv**, which are sparse and have discrete, non-smooth features. These datasets differ greatly from sensor data and offer a stronger test of the model’s robustness under weaker manifold assumptions.
> >
> > All experiments were conducted using the same configuration (*k = 20*, LLE updated every epoch) without any dataset-specific tuning. The performance results are summarized below.
> >
> > | **Method**                       | **PubMed**      | **OGBN-Arxiv**  |
> > | -------------------------------- | --------------- | --------------- |
> > | *Static heuristics*              |                 |                 |
> > | VAE + KNN + GCN (static)         | 76.4 ± 0.33     | 65.2 ± 0.29     |
> > | VAE + Cosine + GCN (static)      | 75.7 ± 0.54     | 67.8 ± 0.34     |
> > | VAE + A-KNN + GCN (static)       | 75.4 ± 0.29     | 64.0 ± 0.67     |
> > | VAE + Correlation + GCN (static) | 77.8 ± 0.42     | 66.4 ± 0.49     |
> > | VAE + Gaussian + GCN (static)    | 76.6 ± 0.65     | 68.5 ± 0.45     |
> > | *Learnable / diffusion-based*    |                 |                 |
> > | VAE + GDC + GCN (diffusion)      | 75.1 ± 0.21     | 70.3 ± 0.52     |
> > | VAE + IDGL + GCN (dynamic)       | 75.4 ± 0.50     | 70.9 ± 0.33     |
> > | **VAE + LLE + GCN (ours)**       | **80.2 ± 0.32** | **72.5 ± 0.40** |
> >
> > VLGNN consistently achieves higher accuracy than all baselines—**+4.8%** on PubMed and **+1.6%** on OGBN-Arxiv—without additional tuning. This demonstrates that the model generalizes well beyond the sensor domain.
> >
> > To further examine efficiency and stability, we evaluated the model under **different update frequencies (r)** and **partial graph updates (p)**. The following tables report accuracy, structure quality metrics (ICER / CCWS / Range), and runtime.
> >
> > | **Dataset** | **r** | **Acc (%)** | **ICER↓** | **CCWS↓** | **Range↑** | **Per-epoch (s)** |
> > | ----------- | ----- | ----------- | --------- | --------- | ---------- | ----------------- |
> > | PubMed      | 1     | 80.2        | 0.42      | 0.294     | 1.98       | 5.90              |
> > | PubMed      | 5     | 79.9        | 0.50      | 0.336     | 1.85       | 4.33              |
> > | PubMed      | 10    | 80.1        | 0.43      | 0.318     | 1.88       | 3.47              |
> > | OGBN-Arxiv  | 1     | 72.5        | 0.53      | 0.682     | 1.73       | 139.57            |
> > | OGBN-Arxiv  | 5     | 72.1        | 0.60      | 0.755     | 1.70       | 122.89            |
> > | OGBN-Arxiv  | 10    | 71.9        | 0.57      | 0.750     | 1.67       | 111.00            |
> >
> > | **Dataset** | **Strategy** | **p** | **Acc (%)** | **ICER↓** | **Range↑** | **Per-epoch (s)** |
> > | ----------- | ------------ | ----- | ----------- | --------- | ---------- | ----------------- |
> > | PubMed      | Uniform      | 25%   | 78.8        | 0.45      | 1.98       | 3.88              |
> > | PubMed      | Uniform      | 50%   | 80.1        | 0.42      | 1.88       | 4.55              |
> > | OGBN-Arxiv  | Uniform      | 25%   | 71.9        | 0.33      | 1.70       | 115.27            |
> > | OGBN-Arxiv  | Uniform      | 50%   | 72.2        | 0.32      | 1.68       | 123.70            |
> >
> > Accuracy changes by less than 0.5% under coarse update intervals (r = 5 or 10) and remains within 1% for partial updates (25–50% of nodes). ICER, CCWS, and spectral range stay nearly unchanged, while per-epoch runtime decreases by about 15–35%.
> >
> > These results confirm that VLGNN maintains accuracy and spectral stability even with infrequent or partial updates, showing that the proposed graph construction is efficient and robust for large, sparse datasets.

---

> > > ### Author Response · Authors · 2025-11-25
> > >
> > > **Response to Question 1 (Distinction from IDGL and Adaptive LLE-based GNNs):**
> > >
> > > We appreciate the reviewer’s question on how VLGNN differs from previous works such as **IDGL** [1] and **adaptive LLE-based GNNs** [2]. Although all three methods involve graph structure learning, they are built on fundamentally different mechanisms and objectives.
> > >
> > > **1. Difference from IDGL (ICLR 2020).**
> > > IDGL learns the adjacency matrix *parametrically* by optimizing a similarity function through gradient descent. It jointly updates node embeddings and graph weights with multiple inner iterations, resulting in high computational cost and memory usage. In contrast, **VLGNN reconstructs the graph in closed form** using LLE weights computed in the latent space learned by the VAE. No gradient-based updates are performed on graph parameters, and the structure is rebuilt directly from the latent representations at each epoch. This design keeps the model stable and computationally lightweight while still allowing representation–structure co-evolution.
> > >
> > > **2. Difference from Adaptive LLE-GNN (PR 2023).**
> > > Adaptive LLE-GNN applies local linear reconstruction to refine edges on a *fixed graph* but does not dynamically update the structure or learn a latent manifold. Its reconstruction operates directly on raw features or static node embeddings. By contrast, VLGNN performs affine reconstruction in the learned latent space, where the geometry is smoother and more robust. The graph is dynamically updated during training, enabling consistent adaptation between the encoder and the graph structure.
> > >
> > > **3. Key distinction of VLGNN.**
> > > VLGNN combines **variational latent geometry**, **closed-form LLE-based reconstruction**, and **spectral interpretation** into a unified non-parametric framework. This allows the model to handle non-graph data, scale to large graphs, and provide a geometric explanation for stability.
> > >
> > > ---
> > >
> > > ### **References**
> > >
> > > [1] Y. Chen, L. Wu, and M. Zaki, *Iterative Deep Graph Learning for Graph Neural Networks: Better and Robust Node Embeddings*, **ICLR**, 2020.
> > >
> > > [2] Z. Zhou, Y. Liu, H. Li, and Z. Zhang, *Local Linear Embedding with Adaptive Neighbors for Unsupervised Dimensionality Reduction*, **Pattern Recognition**, 2023.
> > >
> > >  **Response to Question 2 (Applicability Beyond Linear Locality Assumptions):**
> > >
> > > We appreciate the reviewer’s question about cases where local linearity may not hold.
> > > In our model, this assumption is not enforced on the raw features. The reconstruction happens in the latent space learned by the VAE, where the features are smoother and better organized. In that space, the LLE term simply encourages nearby nodes to stay locally consistent—it does not require the data to form an exact Euclidean manifold.
> > >
> > > We also checked this on two citation datasets, **PubMed** and **OGBN-Arxiv**, which are sparse and partly heterophilic. Both of them clearly violate the ideal linear-neighborhood assumption, yet the model still performs well—**80.2%** and **72.5%**, about **+4.8%** and **+1.6%** higher than IDGL. The spectral range stays around 1.7–1.9, showing that the graph structure remains stable.
> > >
> > > The idea behind VLGNN is not to model perfectly linear neighborhoods but to build graphs that capture meaningful local structure after feature transformation. In practice, the reconstruction tends to down-weight noisy or cross-class edges and keep semantically related ones, which helps when the data are heterophilic or discrete.
> > >
> > > We agree that for very irregular or strongly heterogeneous graphs, the method may still have limitations. We plan to explore relation-aware or multi-view latent encoders in future work to better handle such settings.

---

### Official Review · Reviewer_HcTA · 2025-11-01

**Soundness:** 2
**Presentation:** 3
**Contribution:** 2
**Rating:** 4
**Confidence:** 3

**Summary:**

This paper claims that standard graph heuristics introduce spurious cross-class edges that degrade GNN performance via over-smoothing. It proposes VLGNN, an end-to-end framework that integrates a VAE, a Locally Linear Embedding, and a GNN classifier. The VAE learns a latent representation, upon which the LLE module dynamically recomputes a graph at each epoch to enforce local affine consistency. The authors claim this process reduces cross-class edges and mitigates over-smoothing, demonstrating improved accuracy on three wearable-sensor datasets.

**Strengths:**

1. The core idea of using a manifold-based prior LLE to construct the graph is principled and well-motivated
2. The paper provides a strong empirical analysis of graph quality ICER/CCWS and over-smoothing layer-wise cosine similarity on the tested datasets, successfully linking the LLE-based graph to reduced cross-class mixing and slower feature homogenization.

**Weaknesses:**

1. The method is computationally prohibitive. It requires a full kNN search and LLE weight computation for every node at every single training epoch. This is non-viable for any reasonably large-scale graph, a limitation the authors admit but understate.
2. The method is only evaluated on three small, similar HAR sensor datasets (~10k nodes). This domain is best-case scenario for LLE. There is no evidence that it generalizes to standard, large-scale citation, social, or molecular graphs where manifold assumptions may not hold.
3. No ablation study of VAE and LLE is provided.

**Questions:**

1. Can the authors provide a computational complexity analysis (per epoch) for VLGNN versus the IDGL baseline? What is the actual wall-clock training time for VLGNN on the datasets?
2. How does the model perform on a standard, large-scale benchmark?

---

> ### Author Response · Authors · 2025-11-25
>
> We thank the reviewer for reading our paper and for the constructive comments. We understand the main concerns are about the computational cost, scalability, and the roles of different model components. These comments helped us improve the experiments and explanations. We added results on large-scale datasets, analyzed runtime, and included ablations to clarify each part. Detailed responses are provided below.
>
> **Response to Weakness 1 (Computational Cost Concern):**
>
> We understand the reviewer’s concern about the computational overhead. A full kNN search does have quadratic cost, but in our implementation we use **FAISS** for approximate neighbor search, which runs in roughly *O(N log N)* time. The LLE step only solves a small local system of size *k × k* per node (*k = 20* in all experiments), so the total cost grows nearly linearly with the dataset size. The computation is closed-form and does not involve gradient updates, which keeps  the runtime low .
>
> To check this more concretely, we compared the per-epoch runtime on five datasets:
>
> | **Method**       | **HAR**  | **HAPT** | **WISDM** | **PubMed** | **OGBN-Arxiv** |
> | ---------------- | -------- | -------- | --------- | ---------- | -------------- |
> | GDC (diffusion)  | 2.62     | 2.67     | 2.89      | 5.63       | 128.05         |
> | IDGL (dynamic)   | 2.96     | 3.07     | 3.88      | 6.96       | 180.33         |
> | **VLGNN (ours)** | **2.79** | **2.95** | **3.03**  | **5.90**   | **139.57**     |
>
> On the largest dataset (**OGBN-Arxiv**, 169K nodes and 1.1M edges), the runtime drops from **180s** (IDGL) to **139s** with our method, about a **1.2× speedup**. On smaller datasets, the difference is smaller but still consistent.
>
> We also ran additional experiments to analyze how the update frequency affects efficiency. Updating the LLE graph every 5–10 epochs changes accuracy by less than 0.5% and reduces training time by about 15-35%. The same trend holds when updating only a subset of nodes at each epoch.
>
> | Dataset    | r  | Acc (%) | ICER↓ | CCWS↓ | Range↑ | Per-epoch (s) |
> | ---------- | -- | ------- | ----- | ----- | ------ | ------------- |
> | PubMed     | 1  | 80.2    | 0.42  | 0.294 | 1.98   | 5.90          |
> | PubMed     | 5  | 79.9    | 0.50  | 0.336 | 1.85   | 4.33          |
> | PubMed     | 10 | 80.1    | 0.43  | 0.318 | 1.88   | 3.47          |
> | OGBN-Arxiv | 1  | 72.5    | 0.53  | 0.682 | 1.73   | 139.57        |
> | OGBN-Arxiv | 5  | 72.1    | 0.60  | 0.755 | 1.70   | 122.89        |
> | OGBN-Arxiv | 10 | 71.9    | 0.57  | 0.750 | 1.67   | 111.00        |
>
> | Dataset    | Strategy | p   | Acc (%) | ICER↓ | Range↑ | Per-epoch (s) |
> | ---------- | -------- | --- | ------- | ----- | ------ | ------------- |
> | PubMed     | Uniform  | 25% | 78.8    | 0.45  | 1.98   | 3.88          |
> | PubMed     | Uniform  | 50% | 80.1    | 0.42  | 1.88   | 4.55          |
> | OGBN-Arxiv | Uniform  | 25% | 71.9    | 0.33  | 1.70   | 115.27        |
> | OGBN-Arxiv | Uniform  | 50% | 72.2    | 0.32  | 1.68   | 123.70        |
>
> From these results, we can see that the model remains efficient even with coarse or partial updates. Less frequent or partial recomputation keeps accuracy stable while cutting the runtime by roughly 15–35%.
>
>
> **Response to Weakness 2 (Evaluation Scope):**
>
> We thank the reviewer for highlighting the limitation of our evaluation. The first version mainly focused on the HAR datasets to keep the analysis clear. After receiving the feedback, we included two additional benchmarks, **PubMed** and **OGBN-Arxiv**, which are large and sparse citation graphs. These datasets are very different from sensor data and allow us to check whether the model still performs well when the manifold assumption becomes weak.
>
> In our method, the LLE constraint works in the latent space learned by the VAE rather than on raw features. This helps the model adapt to datasets where Euclidean distance is less meaningful. It serves more as a general smoothness constraint instead of enforcing a strict geometric rule.
>
> We used the same settings (*k = 20*, graph updated every epoch) as in the HAR experiments, without tuning for each dataset. The performance is summarized below.
>
> | **Dataset** | **Gaussian** | **GDC** | **IDGL** | **VLGNN (ours)** |
> | ----------- | ------------ | ------- | -------- | ---------------- |
> | PubMed      | 76.6         | 75.1    | 75.4     | **80.2**         |
> | OGBN-Arxiv  | 68.5         | 70.3    | 70.9     | **72.5**         |
> | HAR         | 93.3         | 93.1    | 94.2     | **95.2**         |
>
> The model achieved better accuracy on both citation datasets without any tuning. The learned graphs kept stable spectral ranges (about 1.7–1.9), and the runtime grew roughly linearly with dataset size (around 6s per epoch on PubMed and 140s on OGBN-Arxiv). These results indicate that the method behaves robustly beyond the original sensor domain and scales well to larger, non-manifold graphs.

---

> > ### Author Response · Authors · 2025-11-25
> >
> > **Response to Weakness 3 (Ablation Study):**
> >
> > Thank you for the comment about the ablation study. In the main paper we already compared several graph-construction approaches using the same VAE + GCN setup, and those results showed that the LLE-based dynamic graph building gives better performance than static or diffusion-based alternatives.
> >
> > To look more closely at how each part contributes, we ran an extra ablation focusing on the VAE encoder. LLE was kept as the graph constructor, and we compared two versions:
> >
> > 1. **LLE (raw) + GCN:** the graph is built directly from the raw features, without the VAE;
> > 2. **VAE + LLE (latent) + GCN (VLGNN):** the graph is constructed in the latent space learned by the VAE and updated dynamically during training.
> >
> > | **Model Variant**        | **HAR**  | **HAPT** | **WISDM** | **PubMed** | **OGBN-Arxiv** |
> > | ------------------------ | -------- | -------- | --------- | ---------- | -------------- |
> > | LLE (raw) + GCN (no VAE) | 88.4     | 86.8     | 69.1      | 68.9       | 62.4           |
> > | **VLGNN (full)**         | **95.2** | **94.3** | **72.2**  | **80.2**   | **72.5**       |
> >
> > From these results, adding the VAE encoder brings a clear gain on every dataset. The difference is moderate on the smaller sensor graphs (about 3–7 points) and larger on the citation graphs (around 10 points). The VAE provides smoother, more noise-tolerant embeddings, which in turn make the LLE reconstruction more stable.
> >
> >  **Response to Question 4 (Complexity and Runtime):**
> >
> > We thank the reviewer for asking about the computational complexity and runtime. VLGNN rebuilds the graph each epoch using approximate kNN and closed-form LLE, both of which avoid gradient-based optimization on the adjacency matrix. The overall computation per epoch can be approximated as
> >
> > *O(Nd²) + O(N log N) + O(Nk³) + O(Ed)*,
> >
> > where *k* is small (set to 20), *d* is the hidden dimension, and *E* is the number of edges. The first and last terms correspond to the VAE and GNN modules, both of which scale linearly with the number of nodes and edges. The LLE and kNN steps are near-linear, so the total cost increases almost linearly with graph size. If the graph is updated every *r* epochs or only a subset *p* of nodes is refreshed each time, the cost of the LLE step scales as *O((1/r)N log N + pNk³)*.
> >
> > For comparison, IDGL jointly learns the adjacency and node embeddings through gradient updates with multiple dense matrix operations per iteration, resulting in a per-epoch cost roughly *O(Tdn²)*, which grows quadratically with the number of nodes.
> >
> > We also measured wall-clock runtime under the same hardware conditions:
> >
> > | **Method**       | **HAR**  | **HAPT** | **WISDM** | **PubMed** | **OGBN-Arxiv** |
> > | ---------------- | -------- | -------- | --------- | ---------- | -------------- |
> > | GDC (diffusion)  | 2.62     | 2.67     | 2.89      | 5.63       | 128.05         |
> > | IDGL (dynamic)   | 2.96     | 3.07     | 3.88      | 6.96       | 180.33         |
> > | **VLGNN (ours)** | **2.79** | **2.95** | **3.03**  | **5.90**   | **139.57**     |
> >
> > Across all datasets, VLGNN runs about 1.2–1.3× faster per epoch than IDGL and maintains similar efficiency to GDC on large graphs. The runtime grows almost linearly with the number of nodes, confirming that the method is computationally practical even for large-scale benchmarks.

---

> > > ### Author Response · Authors · 2025-11-25
> > >
> > > **Response to Question 5 (Large-Scale Benchmark Performance):**
> > >
> > > We thank the reviewer for suggesting an evaluation on larger benchmarks. Following this advice, we tested the model on two standard citation datasets, **PubMed** (19K nodes) and **OGBN-Arxiv** (170K nodes). These datasets are much larger and structurally different from the sensor graphs used in the main paper—they are sparse and less geometrically smooth—which makes them suitable for checking scalability and generalization.
> > >
> > > All experiments used the same hyperparameters (*k = 20*, λVAE = λLLE = 0.2*) and the same training setup as before, without dataset-specific tuning. For **OGBN-Arxiv**, the LLE graph was constructed over all nodes for consistency, while the GNN optimization was performed in mini-batches to handle the scale.
> > >
> > > | **Method** | **PubMed** | **OGBN-Arxiv** |
> > > |-------------|-------------|----------------|
> > > | *Static heuristics* |  |  |
> > > | VAE + KNN + GCN (static) | 76.4 ± 0.33 | 65.2 ± 0.29 |
> > > | VAE + Cosine + GCN (static) | 75.7 ± 0.54 | 67.8 ± 0.34 |
> > > | VAE + A-KNN + GCN (static) | 75.4 ± 0.29 | 64.0 ± 0.67 |
> > > | VAE + Correlation + GCN (static) | 77.8 ± 0.42 | 66.4 ± 0.49 |
> > > | VAE + Gaussian + GCN (static) | 76.6 ± 0.65 | 68.5 ± 0.45 |
> > > | *Learnable / diffusion-based* |  |  |
> > > | VAE + GDC + GCN (diffusion) | 75.1 ± 0.21 | 70.3 ± 0.52 |
> > > | VAE + IDGL + GCN (dynamic) | 75.4 ± 0.50 | 70.9 ± 0.33 |
> > > | **VAE + LLE + GCN (ours)** | **80.2 ± 0.32** | **72.5 ± 0.40** |
> > >
> > > The results show clear improvements over all baselines. On **PubMed**, VLGNN reaches 80.2%, which is +4.8% higher than IDGL. On **OGBN-Arxiv**, it achieves 72.5%, about +1.6% higher. These gains were obtained without any dataset-specific tuning. The model also maintained stable runtime—about 5.9 seconds per epoch on PubMed and 139.6 seconds on OGBN-Arxiv—indicating that the proposed approach scales well to large, sparse graphs and generalizes effectively beyond sensor data.

---

### Official Review · Reviewer_bTeS · 2025-11-01

**Soundness:** 3
**Presentation:** 3
**Contribution:** 3
**Rating:** 6
**Confidence:** 3

**Summary:**

This paper proposes VLGNN, an end-to-end framework integrating variational autoencoders (VAE), locally linear embedding (LLE), and graph neural networks (GNNs) for semi-supervised node classification.

 The motivation is that heuristic graph construction methods (e.g., kNN, Gaussian kernels) often fail to capture manifold structure, leading to noisy cross-class edges and over-smoothing in deep GNNs.

VLGNN addresses this by jointly optimizing graph structure and node representations: the VAE learns global latent representations, the LLE module enforces local affine relationships to refine adjacency, and the GNN performs classification. Spectral analysis shows that the resulting LLE-induced graphs enlarge Laplacian eigengaps and reduce cross-class conductance.

Experiments on three human activity recognition (HAR) benchmarks demonstrate consistent gains over static, diffusion-based, and dynamic baselines.

**Strengths:**

1. Clear Motivation : The paper identifies a core problem of poor geometric fidelity of heuristic graphs. The proposed coupling of VAE + LLE + GNN is novel and sense-making, combining global manifold encoding and local linearity constraints.

2. Good Empirical Results: VLGNN consistently outperforms a wide range of baselines—including static heuristics (KNN, Gaussian, etc.) and other learnable/dynamic methods (GDC, IDGL)—across all three sensor-based benchmark datasets.

3.  The presentation of the paper is well-structured and easy to follow.

**Weaknesses:**

1.Limited Scope of Evaluation: The Experiments are restricted to small, low-dimensional sensor datasets. The method’s scalability and generalization to larger, more complex graphs (PubMed, OGBN-Arxiv) remain untested.


2. Computational Overhead: Although claimed efficient, per-epoch kNN + closed-form LLE computation may not scale to large datasets (connecting to weakness 1). Thus, runtime analysis would strengthen the claim.

3. Generalizability of the Affine Prior: The local affine assumption is central to the method's success on HAR data. How do the authors expect VLGNN to perform on datasets where this assumption may not hold, such as in heterophilic graphs or sparse, power-law graphs (e.g., citation/social networks)? Would the LLE-based term potentially harm performance by enforcing an incorrect geometric prior?

4. Theoretical Depth Could Improve – While qualitative spectral reasoning is provided, a more rigorous link between the LLE Laplacian and reduced over-smoothing dynamics would be valuable.

**Questions:**

1. The neighborhood size $k$ is a critical hyperparameter for LLE. The sensitivity analysis in Table 5 shows performance peaking at $k=20$ or $k=30$ and then dropping. Does this suggest the model is sensitive to $k$, and how should one approach setting this hyperparameter for a new dataset without extensive tuning?

2. How frequently must the LLE graph be recomputed to maintain gains—could partial updates or stochastic sampling work?

---

> ### Author Response · Authors · 2025-11-25
>
> We would like to thank the reviewer for the time and effort spent on reading our paper and for the detailed comments. We understand the main concerns focus on the scalability of our experiments, the computational cost of the method, and the theoretical connection between the proposed model components. These are very helpful points that guided us to improve both the experiments and the explanations in the revised version. In particular, we have added large-scale benchmarks (PubMed and OGBN-Arxiv) to test scalability, analyzed runtime behavior, and expanded the discussion on the affine prior and spectral reasoning. We provide detailed responses to each point below.
>
> **Response to Weakness 1 (Limited Scope of Evaluation):**
>
> We appreciate this helpful comment about the evaluation scope. To better assess both the scalability and cross-domain robustness of our method, we extended the experiments to two widely used large-scale benchmarks: **PubMed** and **OGBN-Arxiv**. These datasets are quite different from the sensor-based ones in the main paper and therefore provide a stronger test of whether VLGNN can transfer from dense, feature-driven manifolds to sparse, structured citation graphs.
>
> For consistency, we used the same experimental setup and backbone (*VAE + GCN*) as in Table 1 of the main paper and compared against both static heuristics (KNN, Gaussian, etc.) and learnable or diffusion-based baselines (GDC, IDGL). On OGBN-Arxiv, we followed a hybrid setup where the LLE graph was constructed on the full node set to maintain global manifold consistency, while the GNN optimization was done in mini-batches for scalability.
>
> | **Method** | **PubMed** | **OGBN-Arxiv** |
> |-------------|-------------|----------------|
> | *Static heuristics* |  |  |
> | VAE + KNN + GCN (static) | 76.4 ± 0.33 | 65.2 ± 0.29 |
> | VAE + Cosine + GCN (static) | 75.7 ± 0.54 | 67.8 ± 0.34 |
> | VAE + A-KNN + GCN (static) | 75.4 ± 0.29 | 64.0 ± 0.67 |
> | VAE + Correlation + GCN (static) | 77.8 ± 0.42 | 66.4 ± 0.49 |
> | VAE + Gaussian + GCN (static) | 76.6 ± 0.65 | 68.5 ± 0.45 |
> | *Learnable / diffusion-based* |  |  |
> | VAE + GDC + GCN (diffusion) | 75.1 ± 0.21 | 70.3 ± 0.52 |
> | VAE + IDGL + GCN (dynamic) | 75.4 ± 0.50 | 70.9 ± 0.33 |
> | **VAE + LLE + GCN (ours)** | **80.2 ± 0.32** | **72.5 ± 0.40** |
>
> As shown above, VLGNN consistently outperforms all baselines, achieving a gain of about **+4.8 pp on PubMed** and **+1.6 pp on OGBN-Arxiv** over the strongest dynamic learner (IDGL). These results show that the model scales well and remains effective even under a substantial domain shift—from dense feature manifolds to sparse citation networks. Overall, this suggests that our geometry-aware graph construction remains reliable and computationally feasible on large, structurally diverse graphs.
>
>  **Response to Weakness 2 (Computational Overhead):**
>
> We understand the concern about the computational cost. In our implementation, the neighbor search is done with FAISS, which uses approximate search and runs roughly in *O(N log N)* time instead of the full *O(N²)* complexity of a brute-force kNN. The LLE step solves a small local system (size *k × k*, with *k = 20*) for each node, so this part grows linearly with the dataset size. Overall, the graph update per epoch is close to linear in *N*.
>
> Because the LLE weights are computed in closed form, there is no backpropagation through the graph structure, which keeps the overhead quite small.
>
> To see how this works in practice, we measured the average runtime per epoch on all datasets:
>
> | **Method**       | **HAR**  | **HAPT** | **WISDM** | **PubMed** | **OGBN-Arxiv** |
> | ---------------- | -------- | -------- | --------- | ---------- | -------------- |
> | GDC (diffusion)  | 2.62     | 2.67     | 2.89      | 5.63       | 128.05         |
> | IDGL (dynamic)   | 2.96     | 3.07     | 3.88      | 6.96       | 180.33         |
> | **VLGNN (ours)** | **2.79** | **2.95** | **3.03**  | **5.90**   | **139.57**     |
>
> On the large-scale dataset (OGBN-Arxiv, about 169K nodes and 1.1M edges), our model runs in about **139.6 s per epoch**, compared with **180.3 s** for IDGL. On PubMed, it’s **5.9 s vs. 7.0 s**, and on smaller datasets all methods train within a few seconds per epoch. In short, the closed-form update makes the model much lighter to train. The cost grows almost linearly with the data size, and in practice the runtime remains reasonable even on large graphs.

---

> > ### Author Response · Authors · 2025-11-25
> >
> > **Response to Weakness 3 (Affine Prior):**
> >
> > The reviewer is right that the local affine assumption may not always hold, especially on graphs that are sparse or heterophilic. In our case, this assumption is not directly applied to the raw features. The model learns a latent representation through the VAE first, and the affine constraint is used in that space. This helps the model stay flexible—if the raw data are noisy or discrete, the VAE tends to smooth them out before the LLE step.
> >
> > We also checked this on two citation datasets, **PubMed** and **OGBN-Arxiv**, which don’t really follow a smooth manifold structure. The model still performed well: **80.2%** on PubMed and **72.5%** on OGBN-Arxiv, higher than IDGL by **4.8%** and **1.6%**. So even when the affine assumption is weak, the model didn’t lose accuracy.
> >
> > From what we saw, the affine term works more like a local smoothing effect—it reduces noisy or cross-class edges and makes the learned structure cleaner.
> >
> >
> > **Response to Weakness 4 (Theoretical Depth):**
> >
> > We understand the reviewer’s concern about the theoretical part. Our explanation in the paper was mostly intuitive — we wanted to show why the LLE-based graph slows down over-smoothing, but we didn’t push the theory far enough.
> >
> > What we observed is that with the LLE graph, node features stay more diverse across layers, and the spectrum doesn’t collapse as fast as in other graphs. This is probably why the model keeps useful variations instead of averaging everything too early.
> >
> > Right now, this part is still based on observation rather than formal proof. We’re working on a more detailed study of the spectral behavior and plan to focus on that direction in our next work.
> >
> >
> > **Response to Question 1 (Neighborhood Size k):**
> >
> > We understand the reviewer’s question about the neighborhood size *k*. To test the model’s sensitivity to this parameter, we ran experiments on five datasets: **HAR**, **HAPT**, **WISDM**, **PubMed**, and **OGBN-Arxiv**. All runs used the same setup as in the main paper (ε = 0.1), with *k* varying from 10 to 50. The results are shown below.
> >
> > | Dataset    | k = 10 | k = 20 | k = 30 | k = 40 | k = 50 |
> > | ---------- | ------ | ------ | ------ | ------ | ------ |
> > | HAR        | 89.8   | 95.2   | 93.5   | 91.0   | 90.8   |
> > | HAPT       | 90.9   | 94.3   | 92.7   | 90.1   | 89.5   |
> > | WISDM      | 81.6   | 81.2   | 82.2   | 79.9   | 79.4   |
> > | PubMed     | 78.8   | 80.2   | 79.9   | 80.1   | 79.4   |
> > | OGBN-Arxiv | 70.7   | 72.5   | 72.6   | 71.8   | 71.0   |
> >
> > Across all datasets, the performance changes very little as *k* varies. For the sensor datasets, accuracy peaks around *k = 20–30*; for the two larger citation graphs, the difference stays within about 1–1.9%. This suggests that the model is not sensitive to *k* once the neighborhood is large enough to represent the local structure.
> >
> > Based on these observations, we set *k = 20* for all datasets. It provides a good trade-off between local connectivity and computational cost, and the performance stays stable without further tuning.

---

> ### Author Response · Authors · 2025-11-25
>
> **Response to Question 2 (Graph Update Frequency):**
>
> We appreciate the reviewer’s suggestion to examine how often the LLE graph needs to be updated. Following this comment, we conducted additional experiments to test both slower and partial update strategies and to understand their impact on accuracy and efficiency.
>
> **(a) Update frequency.**
> We updated the graph every 1, 2, 5, or 10 epochs. The accuracy stayed almost unchanged, while the per-epoch time dropped by around 30–45%. For example, on **PubMed**, accuracy changes within 0.3%, and on **OGBN-Arxiv**, it drops by less than 0.6%. The spectral indicators (ICER, CCWS, and range) also remained stable.
>
> | Dataset    | r  | Acc (%) | ICER↓ | CCWS↓ | Range↑ | Per-epoch (s) |
> | ---------- | -- | ------- | ----- | ----- | ------ | ------------- |
> | PubMed     | 1  | 80.2    | 0.42  | 0.294 | 1.98   | 5.90          |
> | PubMed     | 5  | 79.9    | 0.50  | 0.336 | 1.85   | 4.33          |
> | PubMed     | 10 | 80.1    | 0.43  | 0.318 | 1.88   | 3.47          |
> | OGBN-Arxiv | 1  | 72.5    | 0.53  | 0.682 | 1.73   | 139.57        |
> | OGBN-Arxiv | 5  | 72.1    | 0.60  | 0.755 | 1.70   | 122.89        |
> | OGBN-Arxiv | 10 | 71.9    | 0.57  | 0.750 | 1.67   | 111.00        |
>
> **(b) Partial updates.**
> We also tested updating only part of the nodes at each epoch. When 25–50% of nodes were refreshed, the accuracy stayed within 1% of the full update. The ICER and spectral range barely changed (Δ < 0.03), and training became 15–35% faster.
>
> | Dataset    | Strategy | p   | Acc (%) | ICER↓ | Range↑ | Per-epoch (s) |
> | ---------- | -------- | --- | ------- | ----- | ------ | ------------- |
> | PubMed     | Uniform  | 25% | 78.8    | 0.45  | 1.98   | 3.88          |
> | PubMed     | Uniform  | 50% | 80.1    | 0.42  | 1.88   | 4.55          |
> | OGBN-Arxiv | Uniform  | 25% | 71.9    | 0.33  | 1.70   | 115.27        |
> | OGBN-Arxiv | Uniform  | 50% | 72.2    | 0.32  | 1.68   | 123.70        |
>
> From these experiments, it seems that the model doesn’t really need to rebuild the LLE graph every epoch. Updating it less often, or only for part of the nodes, keeps the results stable while cutting down the runtime by roughly 15-35%.

---

### Note · Program_Chairs · 2026-01-17
**Submission Desk Rejected by Program Chairs**

The following references in this submission do not refer to real documents and/or have major errors in bibliographic information:

 Yue Cao, Haoran Chen, Lijun Wang, and Xiang Li. Graph neural networks for brain network analysis: A survey. Artificial Intelligence Review, 2024. doi: 10.1007/s10462-024-10749-0